# Sample then Identify: A General Framework for Risk Control and Assessment in Multimodal Large Language Models

**Qingni Wang**[1], **Tiantian Geng**[2,3], **Zhiyuan Wang**[1], **Teng Wang**[2,4], **Bo Fu**[1,*]**Feng Zheng**[2*]

[1]University of Electronic Science and Technology of China
[2]Southern University of Science and Technology
[3]University of Birmingham
[4]The University of Hong Kong
{qingni1031,gengtiantian97,yhzywang,ttengwang}@gmail.com

## Abstract

Multimodal Large Language Models (MLLMs) exhibit promising advancements across various tasks, yet they still encounter significant trustworthiness issues. Prior studies apply Split Conformal Prediction (SCP) in language modeling to construct prediction sets with statistical guarantees. However, these methods typically rely on internal model logits or are restricted to multiple-choice settings, which hampers their generalizability and adaptability in dynamic, open-ended environments. In this paper, we introduce *TRON*, a **t**wo-step framework for **r**isk c**o**ntrol and assessme**n**t, applicable to any MLLM that supports sampling in both open-ended and closed-ended scenarios. *TRON* comprises two main components: (1) a novel conformal score to **sample** response sets of minimum size, and (2) a nonconformity score to **identify** high-quality responses based on self-consistency theory, controlling the error rates by two specific risk levels. Furthermore, we investigate semantic redundancy in prediction sets within open-ended contexts for the first time, leading to a promising evaluation metric for MLLMs based on average set size. Our comprehensive experiments across four Video Question-Answering (VideoQA) datasets utilizing eight MLLMs show that *TRON* achieves desired error rates bounded by two user-specified risk levels. Additionally, deduplicated prediction sets maintain adaptiveness while being more efficient and stable for risk assessment under different risk levels.

## 1 Introduction

In recent years, there has been swift progress in the development of Large Language Models (LLMs) and Multimodal Large Language Models (MLLMs) (Zhao et al., 2023; Wu et al., 2023). MLLMs extend the capabilities of LLMs by integrating and processing information from multiple modalities, including text, vision, and audio (Bai et al., 2024; Zhu et al., 2024). However, MLLMs are proven to exhibit significant drawbacks in trustworthiness, such as hallucination (Rawte et al., 2023; Zhang et al., 2024; Wang et al., 2024a), which results in non-factual information (Ji et al., 2023) and biased generations (Zou et al., 2023). These issues have elicited increasing societal concerns regarding the reliable deployment of foundation models in consumer-facing applications (Qian et al., 2024).

Uncertainty estimation (Ye et al., 2024) provides valuable insights into the trustworthiness of model generations. Prior work performs probabilistic modeling (Li et al., 2024), develops entropy-based measures (Wang et al., 2024c; Duan et al., 2024; Farquhar et al., 2024), and generates multiple responses to analyze the output space (Lin et al., 2024; Wang et al., 2024b). However, these approaches cannot provide guarantees of the error rate, and forcing a measure to predict a single data point is overly restrictive (Cresswell et al., 2024). Split Conformal Prediction (SCP) (Papadopoulos et al., 2002; Angelopoulos & Bates, 2021; Campos et al., 2024) is a promising candidate to tackle these

---

*Corresponding co-authors: fubo@uestc.edu.cn, f.zheng@ieee.org

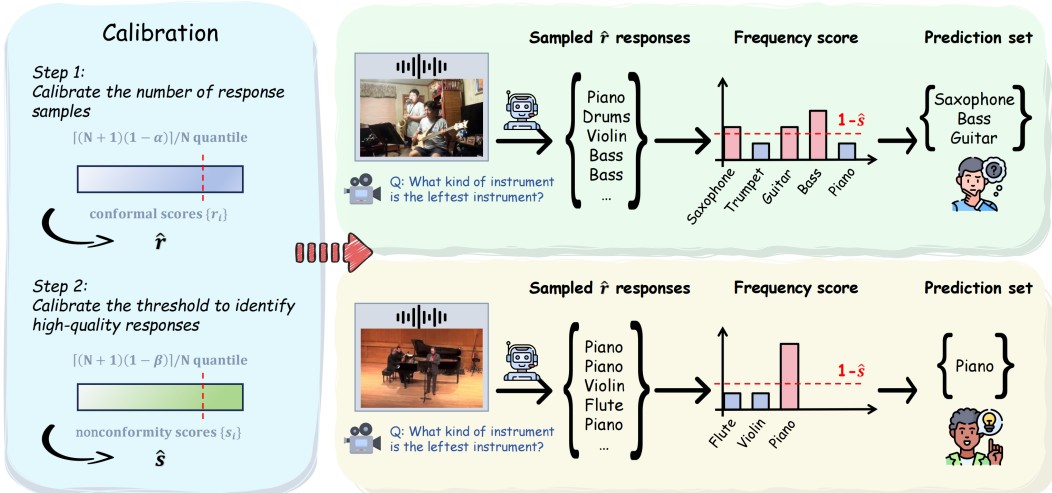

Figure 1: Pipeline of *TRON* illustrated by two open-ended VideoQA examples. Step 1 calibrates the minimum number of responses required to **sample** for each test data point on the calibration set; Step 2 calibrates the threshold to **identify** high-quality responses. A larger prediction set size reflects higher risk (model uncertainty) during the current VideoQA task.

challenges above, which constructs statistically rigorous prediction sets with the hallmark of risk control (Angelopoulos et al., 2024; Farinhas et al., 2024). Furthermore, SCP allows the calibrated set size to vary based on the model's uncertainty regarding a particular input.

SCP has been successfully applied in language modeling (Campos et al., 2024). However, existing methods either (1) modify the original model outputs to ensure factuality with desired probability (Mohri & Hashimoto, 2024; Cherian et al., 2024), potentially leading to uninformative and vague generations, (2) rely on internal token sequence logits (Quach et al., 2024), or (3) are restricted to multiple-choice settings (Kumar et al., 2023; Ye et al., 2024; Kostumov et al., 2024). Adapting SCP for proprietary MLLMs in practical open-ended video question-answering (VideoQA) applications presents several challenges: (a) *Adaptability*. Unlike closed-ended tasks equipped with fixed options covering the correct answers, the output space for each VideoQA task is unbounded, and there may not be an acceptable response within the sampled generations; (b) *Reliability*. Logits or verbalized confidence levels are often miscalibrated, leading to biased prediction sets; (c) *Flexibility*. For some API-only MLLMs, users lack access to internal model information like logits.

In this paper, we aim to tackle these challenges by developing a **t**wo-step framework for **r**isk c**o**ntrol and assessme**n**t (*TRON*). We illustrate the pipeline of *TRON* via two open-ended VideoQA samples in Figure 1. The workflow begins by determining whether acceptable responses exist in the sampled generations, akin to options in closed-ended tasks. To this end, we propose a novel *conformal score* that controls the minimum number of response **samples** required for each calibration data point. For a new data point, we define the number of samples as the approximate $1-\alpha$ quantile of all conformal scores in the calibration set, denoted as $\hat{r}$. Then, the error rate for the candidate set of each test data point failing to encompass an acceptable response is controlled by the risk level $\alpha$ (i.e., $\leq \alpha$).

Once the risk level for acceptable responses being sampled is specified, we evaluate the reliability of each response. Inspired by self-consistency theory (Wang et al., 2022; Li et al., 2022b), we propose using the frequency of each response as a proxy for confidence[1], compensating for the limitations of internal logits information and verbalized confidence scores, and define the *nonconformity score* of each calibration data as one minus the frequency score of any one acceptable response in the candidate set. Then, we specify another risk level $\beta$ and calculate the approximate $1-\beta$ quantile of all nonconformity scores, denoted as $\hat{s}$. The quantile serves as a threshold to **identify** high-quality responses within the candidate set of each test data point. Finally, we guarantee that the miscoverage

---

[1]Response frequency positively correlates with average real generative probability. As the frequency increases, there is a corresponding growing level of confidence in the model's responses (Su et al., 2024).

rate of acceptable responses in the calibrated prediction sets is statistically controlled by two risk levels, $\alpha$ and $\beta$ (i.e., $\leq \alpha + \beta - \alpha\beta$), while the set size reflects the model uncertainty to each query.

We evaluate our framework on two closed-ended VideoQA datasets (e.g., Video-MME (Fu et al., 2024) and NExT-QA (Xiao et al., 2021)), and two open-ended VideoQA datasets (e.g., MUSIC-AVQA (Li et al., 2022a) and MSVD (Chen & Dolan, 2011)), utilizing five open-source MLLMs(e.g., VideoLLaMA (Zhang et al., 2023), PandaGPT (Su et al., 2023), and NExT-GPT (Wu et al., 2024)), and three closed-source MLLMs(e.g., Gemini (Reid et al., 2024) and GPT-4o mini (OpenAI, 2023)). Experimental results demonstrate that *TRON* guarantees the error rates in both steps across various user-specified risk levels. Additionally, the deduplicated average set size provides stable uncertainty estimation of MLLMs across various risk levels in open-ended VideoQA tasks, complementing the accuracy metric for more comprehensive evaluations. Furthermore, we investigate other black-box measurements in the development of the nonconformity score, resulting in more efficient predictions and consistently rigorous guarantees. To summarize, our contributions are the following:

- We introduce a general framework for risk control and assessment, applicable to any MLLM that supports sampling for both open-ended and closed-ended VideoQA tasks.

- We propose a novel conformal score to derive the minimum sample size that bounds the miscoverage rate, extending SCP to open-ended contexts, and then develop the nonconformity score based on black-box measures, which results in rigorous guarantees as well.

- We explore the impact of semantic redundancy on risk assessment leveraging prediction set in open-ended settings, and disclose that the deduplicated set size provides more stable uncertainty estimation across various risk levels, which serves as a promising evaluation metric for MLLMs in VideoQA tasks.

## 2 RELATED WORK

**Split Conformal Prediction and Risk Control.** Our work extends the variants of the SCP framework for correctness (coverage) guarantees and risk control in language modeling. Existing methods either modify model generations and achieve conformal factuality guarantees (Mohri & Hashimoto, 2024; Cherian et al., 2024) or guarantee the ground-truth coverage in Multiple-Choice Question-Answering (MCQA) tasks (Kumar et al., 2023; Ye et al., 2024; Kostumov et al., 2024). The previous approaches may result in vague and uninformative responses, constrained by the correctness of the original generation, while the latter frameworks are limited to closed-ended settings. In open-ended scenarios, (Quach et al., 2024) develops a stopping rule with slightly relaxed coverage criteria utilizing the internal logits information, restricting the applicability to API-only models, whereas (Su et al., 2024) leverages black-box uncertainty but fails to align the nonconformity score with correctness, leading to theoretically loose guarantees. Our approach addresses these gaps by introducing a novel two-step framework that first derives the minimum number of response samples to approximate a "closed-ended" condition, where there is at least one acceptable option provided, followed by identifying reliable responses based on self-consistency theory, which features (1) rigor, controlled by two user-specified risk levels, $\alpha$ and $\beta$, and (2) versatility, applicable to any MLLM that supports sampling in both closed-ended and open-ended settings.

**Risk Assessment and Uncertainty Estimation.** Recent work performs probabilistic modeling (Li et al., 2024), develops entropy-based uncertainty measures (Wang et al., 2024c; Duan et al., 2024; Farquhar et al., 2024), and generate multiple responses to analyze the output space (Lin et al., 2024; Wang et al., 2024b). However, these approaches are heuristic, and forcing a measure to predict a single data point is overly restrictive (Cresswell et al., 2024). (Ye et al., 2024) and (Kostumov et al., 2024) apply SCP to MCQA tasks, creating conformal sets based on a measure of uncertainty (risk), where the set size represents the confidence of the foundational model regarding a particular input, with larger sets indicating higher risk. In open-ended scenarios, there may be redundant responses within the prediction sets due to the randomness of sampling (Quach et al., 2024). Our work explores the performance of set size on risk assessment before and after deduplication and observes that the original set size becomes unstable in uncertainty estimation across different MLLMs as the error rate changes, while the calibrated set size reliably evaluates MLLMs across various risk levels.

## 3 METHODOLOGY

In this section, we introduce how our framework extends SCP techniques to open-ended VideoQA tasks based on a novel conformal score for sampling responses. In addition, to address the limitation of internal logits on API-only MLLMs, we develop the nonconformity score for identifying high-quality responses based on black-box measures.

### 3.1 PRELIMINARIES

Following standard SCP for risk control (Angelopoulos & Bates, 2021; Angelopoulos et al., 2024), we assume a calibration dataset $\mathcal{D}_{cal} = \{(X_i, Y_i)\}_{i=1}^N$ of $N$ query-answer pairs and an independent and identically distributed (i.i.d.)[2] test point $(x_{test}, y_{test})$. Our goal of risk control is to construct prediction sets that are calibrated to bound the correctness miscoverage rate

$$\mathbb{P}\{y_{test} \notin \mathcal{C}(x_{test})\} \leq \varepsilon, \tag{1}$$

where $\varepsilon$ is the user-specified risk level, and $\mathcal{C}$ is a function formed utilizing $\mathcal{D}_{cal}$. Note that the risk level is marginal (average) over the draw of both the calibration and test data points.

For each query of calibration data $X_i$, we generate $M_i$ response samples $\left\{\hat{y}_m^{(i)}\right\}_{m=1}^{M_i}$ and demand that there is at least one correct response in the candidate set (i.e., $Y_i \in \left\{\hat{y}_m^{(i)}\right\}_{m=1}^{M_i}$). For the test point, we also generate $M_{test}$ response samples $\left\{\hat{y}_m^{(test)}\right\}_{m=1}^{M_{test}}$. Note that we are currently unsure if the candidate set of test data encompasses acceptable responses.

### 3.2 CONFORMAL RISK CONTROL

**Sampling.** As we construct a prediction set by selecting high-quality responses from the candidate set, we first perform a simple calibration step for the setting of $M_{test}$ to control the risk of $\left\{\hat{y}_m^{(test)}\right\}_{m=1}^{M_{test}}$ failing to encompass acceptable responses, so that we can approximate closed-ended settings with a specific probability. We develop a novel conformal score for each calibration data point, which determines the minimum sampling threshold that ensures $Y_i \in \left\{\hat{y}_m^{(i)}\right\}_{m=1}^{M_i}$

$$r(X_i, Y_i) := \sup\left\{\hat{M}_i : \forall M_i^{'} < M_i, Y_i \notin \left\{\hat{y}_m^{(i)}\right\}_{m=1}^{M_i^{'}}\right\}, \tag{2}$$

and define $\hat{r}$ to be the $\frac{\lceil(N+1)(1-\alpha)\rceil}{N}$ quantile of $r_1, \cdots, r_N$ ($r_i = r(X_i, Y_i)$): $\hat{r} = r_{\lceil(N+1)(1-\alpha)\rceil}$. Then, we set $M_{test} = \hat{r}$ and obtain the upper bound of the error rate

$$\mathbb{P}\left(y_{test} \notin \left\{\hat{y}_m^{(test)}\right\}_{m=1}^{\hat{r}}\right) \leq \alpha. \tag{3}$$

A complete proof of Eq. 3 is presented in Appendix B.1.

**Identification.** Inspired by self-consistency theory (Wang et al., 2022; Li et al., 2022b; Wang et al., 2024b), which states that a repetitively sampled response is viewed as a form of consistency linked to higher confidence in the response, we perform a simple semantic clustering process on the candidate set and obtain the frequency of each response, denoted as $F\left(\hat{y}_m^{(i)}\right)$. For a detailed description of semantic clustering, refer to Appendix C.1. Then, we define the nonconformity score of the $i$-th calibration data point as one minus the frequency of the acceptable response, denoted as $\hat{y}_{ref}^{(i)}$, within the candidate set, which is semantically equivalent to the label $Y_i$ (i.e., $\hat{y}_{ref}^{(i)} \Leftrightarrow Y_i$). $A \Leftrightarrow B$ represents a bidirectional entailment (MacCartney & Manning, 2015; Mohri & Hashimoto, 2024;

---

[2]Split conformal prediction requires that the test data points are drawn from the same distribution as the calibration data points.

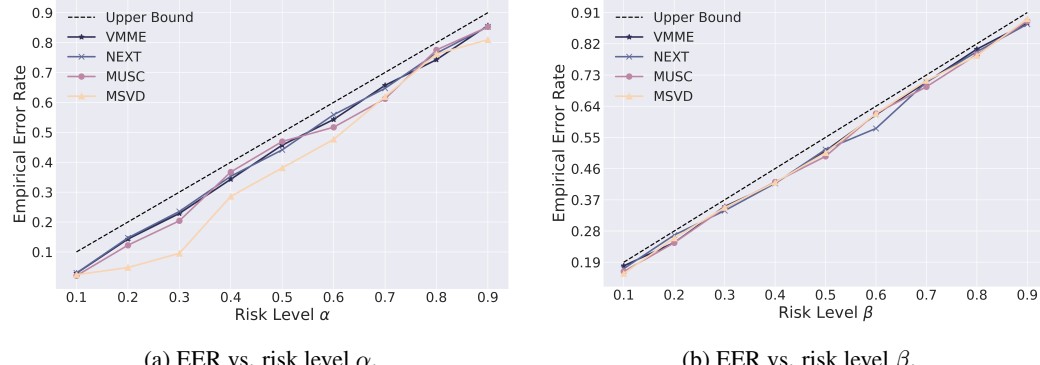

(a) EER vs. risk level $\alpha$.  (b) EER vs. risk level $\beta$.

Figure 2: Results of the EER metric at various risk levels. Each solid line starts at the GPT-4o mini model on the associated dataset. In (a), the upper bound corresponds to various values of the risk level $\alpha$. In (b), we fix the risk level in the first stage at 0.1 (i.e., $\alpha = 0.1$), and the upper bound represents the values of $\varepsilon$ for different values of $\beta$.

Farquhar et al., 2024) between A and B (i.e., equivalence). The specific evaluation for bidirectional entailment is provided in Appendix C.1. In this case, the nonconformity score is formulated as

$$s\left(X_i, Y_i\right) := 1 - F\left(\hat{y}_{ref}^{(i)}\right). \tag{4}$$

Note that $\hat{y}_{ref}^{(i)}$ refers to any acceptable response in the candidate set that is semantically equivalent to the reference answer $Y_i$. Similar to $\hat{r}$, we define $\hat{s}$ to be the $\frac{\lceil(N+1)(1-\beta)\rceil}{N}$ quantile of $s_1, \cdots, s_N$ ($s_i = s\left(X_i, Y_i\right)$): $\hat{s} = s_{\lceil(N+1)(1-\beta)\rceil}$. Then, we construct the prediction set by post-processing the candidate set into a set of responses with low non-conformity scores

$$\mathcal{C}\left(x_{test}\right) := \left\{\hat{y}_{m'}^{(test)} \in \left\{\hat{y}_m^{(test)}\right\}_{m=1}^{\hat{r}} : s\left(x_{test}, \hat{y}_{m'}^{(test)}\right) \le \hat{s}\right\}. \tag{5}$$

Finally, we guarantee the upper bound of the correctness miscoverage rate

$$\mathbb{P}\left\{y_{test} \notin \mathcal{C}\left(x_{test}\right)\right\}$$
$$= \mathbb{P}\left(y_{test} \notin \left\{\hat{y}_m^{(test)}\right\}_{m=1}^{\hat{r}}\right) + \mathbb{P}\left(s_{test} > \hat{s} \mid y_{test} \in \left\{\hat{y}_m^{(test)}\right\}_{m=1}^{\hat{r}}\right) \tag{6}$$
$$\le \varepsilon,$$

where $\varepsilon = \alpha + \beta - \alpha\beta$. The complete proof of Eq. 6 is given in Appendix B.2.

**Extensibility.** $F\left(\cdot\right)$ can be any measure that reflects the reliability of each response. In the case of frequency, an appropriate sample size is necessary to approximate the output distribution and accurately represent the confidence level. We also investigate other measurements in Section 4.4 and evaluate their impact on risk control and assessment and prediction efficiency.

## 4 EMPIRICAL EVALUATIONS

### 4.1 EXPERIMENTAL SETTINGS

**Models.** We employ five open-source MLLMs, including VideoLLaMA-7B (Zhang et al., 2023), VideoLLaMA-13B, PandaGPT-7B (Su et al., 2023), PandaGPT-13B, and NExT-GPT (Wu et al., 2024), and three closed-source MLLMs, including Gemini-1.5-Flash (Reid et al., 2024), Gemini-1.5-Pro, and GPT-4o mini (OpenAI, 2023). Note that we utilize all five open-source MLLMs as if they are API-only MLLMs, i.e., it assumes no access to any internal information of MLLMs.

**Datasets.** We consider two multiple-choice VideoQA datasets: Video-MME (VMME) (Fu et al., 2024) and NExT-QA (NEXT) (Xiao et al., 2021), where we only utilize the multiple-choice section of NEXT, and two open-ended VideoQA datasets: MUSIC-AVQA (MUSC) (Li et al., 2022a) and MSVD (Chen & Dolan, 2011). More details of the dataset utilization can be found in Appendix C.

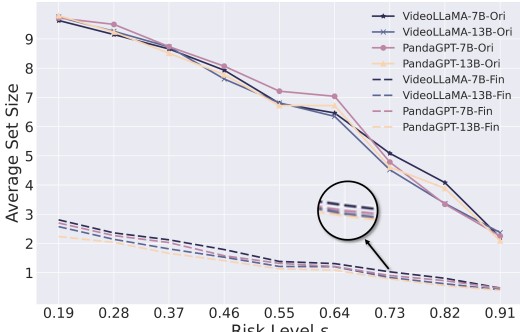

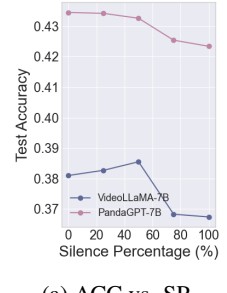

(a) ACC vs. SP.

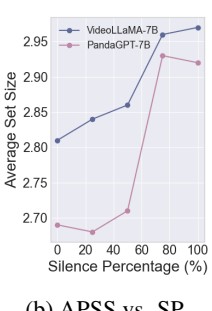

(b) APSS vs. SP.

Figure 3: Comparison of APSS before and after deduplication at different risk levels ($\varepsilon$) across four open-source MLLMs on the VMME dataset.

Figure 4: (a) and (b) show the variation of ACC and APSS across various SPs, utilizing the VideoLLaMA-7B and PandaGPT-7B models on the VMME dataset.

**Evaluation Metrics.** We utilize the Empirical Error Rate (EER) to assess whether we control the risk of miscoverage by the two user-specified risk levels (Angelopoulos & Bates, 2021). Note that in the first stage, miscoverage refers to not sampling the acceptable responses, and this EER is controlled by $\alpha$. In the second stage, miscoverage refers to the calibrated prediction sets failing to cover the acceptable responses, and this EER is bounded by both $\alpha$ and $\beta$. Additionally, we adopt the commonly used Accuracy (ACC) and the Average Prediction Set Size (APSS) (Kostumov et al., 2024; Ye et al., 2024) to evaluate MLLM predictions. More details can be found in Appendix F.

## 4.2 RESULTS FOR RISK CONTROL

Following prior work (Mohri & Hashimoto, 2024; Su et al., 2024; Quach et al., 2024), we randomly split our datasets in half by default, calibrating the sampling number and threshold for identifying high-quality responses on the first half, i.e., the calibration set, and measure the EER (i.e., miscoverage rate) at various risk levels (i.e., upper bound) in two steps on the second half, i.e., the test set. We plot the empirical results in the two steps utilizing the GPT-4o mini model on four datasets in Figure 2a and Figure 2b, and defer similar plots employing other MLLMs to Appendix G.

In the first step, we aim to address the challenge of determining whether acceptable responses are covered by the candidate set in open-ended VideoQA tasks. Figure 2a demonstrates that by deriving the minimum number of responses required to sample for each test data point on the calibration set based on our developed conformal score, we achieve statistically rigorous guarantees on EER. The marginal correctness miscoverage rates by the sampled responses on the test set are consistently bounded by various predetermined risk levels. In this way, we can approximate the fixed options provided in closed-ended tasks through sampling, despite the presence of duplicate options.

In the second step, we attempt to overcome the limitations of previous approaches, which rely on internal model logits or verbalized confidence scores. We specify the risk level in the first step to be 0.1 (i.e., $\alpha = 0.1$), and then measure the correctness miscoverage rates by the prediction sets, which are calibrated by our devised nonconformity score based on self-consistency theory. Figure 2b illustrates the validity of identifying high-quality responses based on their frequency scores with the candidate sets. The miscoverage rates on the calibrated prediction sets are strictly bounded by the risk level $\varepsilon$ ($= \alpha + \beta - \alpha\beta$) at various settings of $\beta$. In this way, we obtain an intuitive confidence level by analyzing the frequency of each response in the candidate set, adapting our framework to API-only MLLMs. In terms of the black-box reliability measurement, we also evaluate the development of nonconformity score based on semantic diversity in Section 4.4.

Empirical evaluations demonstrate that our risk control framework matches our theory in practice, as the correctness miscoverage rates in both steps never exceed the user-specified risk level. Note that EER is marginal (average) over the test set, not conditional to individual test data points.

Table 1: Evaluations of the ACC and APSS metrics utilizing five open-source and three closed-source MLLMs across four open-ended and closed-ended datasets.

| MLLMs | ACC (%) ↑ | | | | APSS ↓ | | | |
|---|---|---|---|---|---|---|---|---|
| | VMME | NEXT | MUSC | MSVD | VMME | NEXT | MUSC | MSVD |
| *Open-source Models* | | | | | | | | |
| VideoLLaMA-7B | 38.10 | 46.06 | 78.70 | 78.29 | 2.81 | 2.68 | 1.97 | 1.63 |
| VideoLLaMA-13B | 45.33 | 47.07 | 79.18 | 78.91 | 2.58 | 2.37 | 1.59 | 1.63 |
| PandaGPT-7B | 43.45 | 50.95 | 81.85 | 77.29 | 2.70 | 2.10 | 2.02 | 1.12 |
| PandaGPT-13B | 48.75 | 58.90 | 82.73 | 86.04 | 2.24 | 2.20 | 2.03 | 2.21 |
| NExT-GPT | 42.64 | 42.59 | 79.84 | 84.37 | 2.43 | 2.45 | 2.20 | 1.70 |
| *Closed-source Models* | | | | | | | | |
| Gemini-1.5-Flash | 72.31 | 70.80 | 85.58 | 87.14 | 1.22 | 1.17 | 1.55 | 1.08 |
| Gemini-1.5-Pro | 73.11 | 76.29 | 85.16 | 86.67 | 1.15 | 1.15 | 1.59 | 1.10 |
| GPT-4o mini | 73.26 | 82.50 | 86.27 | 85.71 | 1.14 | 1.08 | 1.43 | 1.21 |

## 4.3 RESULTS FOR RISK ASSESSMENT

Conformal sets are constructed adaptively to each particular input (Angelopoulos & Bates, 2021), so that the set size gives us an indicator of the model's uncertainty (or the risk in the current decision-making). Prior work (Ye et al., 2024; Kostumov et al., 2024) utilize APSS to benchmark LLMs and Vision-Language Models (VLMs) through risk assessment in MCQA tasks. In contrast to closed-ended scenarios, where MLLMs are provided a predetermined set of options that encompasses the correct answer and produce a prediction set without internal duplicates, open-ended VideoQA tasks feature their unfixed and unbounded output spaces. Consequently, duplicate responses are inevitable during the sampling process, which leads to semantic redundancy[3] in the prediction set after filtering for high-quality responses based on the nonconformity score.

In this section, we investigate the impact of semantic redundancy in the prediction set when employing the APSS metric to evaluate MLLMs, to address the research gap in open-ended settings. We fix the risk level in the first stage at 0.1 (i.e., $\alpha = 0.1$) and measure the APSSs for different values of $\varepsilon$ ($= \alpha + \beta - \alpha\beta$), utilizing four open-source MLLMs on the VMME dataset. As demonstrated in Figure 3, the original APSS is sensitive to the error rate, leading to ambiguity in the evaluation of MLLMs at different risk levels, while after removing semantically redundant responses, the final APSS can consistently decrease as the error rate lowers, and accurately distinguish between foundational models of varying performance at different risk levels. Additionally, the post-processed prediction sets result in more efficient decision-making. Note that we still utilize bidirectional entailment detailed in Appendix C to determine the semantic equivalence between two responses.

Given the adaptiveness and stability of the deduplicated APSS, we propose utilizing uncertainty/risk, represented by APSS, to evaluate MLLMs in open-ended VideoQA tasks for the first time. We specify the risk level $\varepsilon$ to be 0.19 ($\alpha = \beta = 0.1$) and evaluate the ACC and APSS metrics across four datasets utilizing eight MLLMs. Intuitively, MLLMs with higher accuracy should exhibit lower risk when processing a given VideoQA dataset. However, as shown in Table 1, higher ACC does not necessarily correspond to lower uncertainty and risk. For instance, although PandaGPT-13B achieves an ACC of 86.04% on the MSVD dataset, higher than GPT-4o mini, which has an ACC of 85.71%, PandaGPT-13B exhibits a higher APSS, indicating greater uncertainty compared to GPT-4o mini. This demonstrates that even with higher ACC, uncertainty, measured by APSS, may not decrease. Similarly, VideoLLaMA-13B and PandaGPT-13B both demonstrate a positive correlation between ACC and APSS on the MSVD dataset. These findings highlight the importance of considering uncertainty alongside ACC when evaluating MLLMs.

Unlike prior work (Kostumov et al., 2024) that evaluates MLLMs on vision-language QA datasets, we investigate the impact of the audio modality on MLLM performance in open-ended VideoQA tasks. Specifically, we extract audio information from the original video, apply varying levels of Silence Percentage (SP), defined as the proportion of randomly muted audio segments in a video, and then provide the processed audio, along with the corresponding visual and textual information,

---

[3]In the open-ended VideoQA task, sampling multiple responses from MLLMs can result in the candidate set containing identical answers or answers that are syntactically or lexically distinct but semantically equivalent.

to MLLMs. As illustrated in Figure 4, incorporating additional modality information enhances the ACC of MLLMs to some extent. As the SP increases, we observe significant changes in model uncertainty through the APSS metric, showcasing that introducing audio modality enhances the confidence level of MLLMs' decision-making. In light of the observations mentioned above, we hope that our framework can assist in evaluating the performance of MLLMs in VideoQA tasks through risk assessment, while providing guarantees, thereby complementing the accuracy metric for a more comprehensive analysis.

## 4.4 SENSITIVITY ANALYSES

**Splitting Ratio of Calibration and Test Set.** As previously discussed, the calibration set is essentially a set of sufficiently general observation samples, from which we derive the number of response samples required for each test data point and the threshold for identifying high-quality responses in the candidate set based on specific statistical criteria. In this section, we explore the impact of the ratio of sample sizes between the calibration set and the test set on the final performance, specifically how many resources are needed for our framework to achieve risk guarantees on the test set. In previous work, we set a ratio of 0.5 by default. Here, we evaluate the EER when the ratio is set to 0.3 and 0.1. As shown in Figure 5, when $\alpha$ and $\beta$ are set to 0.1, resulting in an overall risk level (i.e., $\varepsilon$) of 0.19, we achieve consistently strict guarantees of the miscoverage rates across three split ratios utilizing three closed-source MLLMs on the MUSC dataset, which demonstrates the efficiency and stability of our framework for risk control in practical open-ended VideoQA applications.

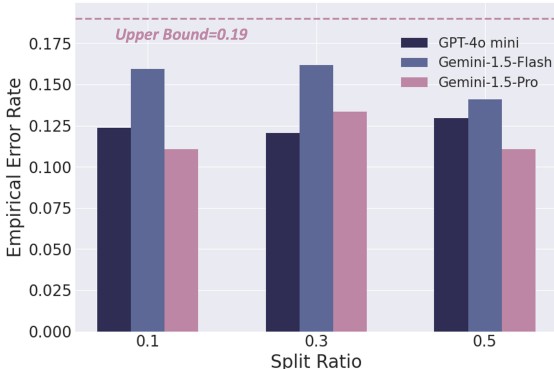

Figure 5: EERs on the MUSC dataset utilizing three closed-source MLLMs when varying the split ratio of the calibration and test set. The risk levels $\alpha$ and $\beta$ are both set to be 0.1.

Table 2: Comparison of frequency and semantic diversity on the VMME datasets employing four open-source MLLMs. The risk levels $\alpha$ and $\beta$ are both set to be 0.1.

| MLLMs | APSS | | EER | |
|---|---|---|---|---|
| | **Frequency** | **Semantic Diversity** | **Frequency** | **Semantic Diversity** |
| VideoLLaMA-7B | 2.81 | 2.74 | 0.1793 | 0.1766 |
| VideoLLaMA-13B | 2.58 | 2.52 | 0.1767 | 0.1744 |
| PandaGPT-7B | 2.70 | 2.42 | 0.1812 | 0.1675 |
| PandaGPT-13B | 2.24 | 2.20 | 0.1783 | 0.1674 |

**Reliability Measurement.** As discussed in Section 3.2, $F(\cdot)$ in the second step can be any measure that evaluates the reliability of each response within the candidate set. To address the issues of logit access utilizing API-only MLLMs and the overconfidence in the verbalized reliability scores, we leverage the most intuitive frequency score based on self-consistency theory. In this section, we utilize the semantic diversity, formulated as $\sum_{\hat{y}_j^{(i)} \neq \hat{y}_m^{(i)}} S\left(\hat{y}_j^{(i)}, \hat{y}_m^{(i)}\right) F\left(\hat{y}_j^{(i)}\right)$, as the reliability score of the $m$-th response with the $i$-th candidate set. We set the risk levels $\alpha$ and $\beta$ to be 0.1 (i.e., $\varepsilon = 0.19$), and evaluate the APSS and EER metrics on the VMME dataset utilizing four open-source

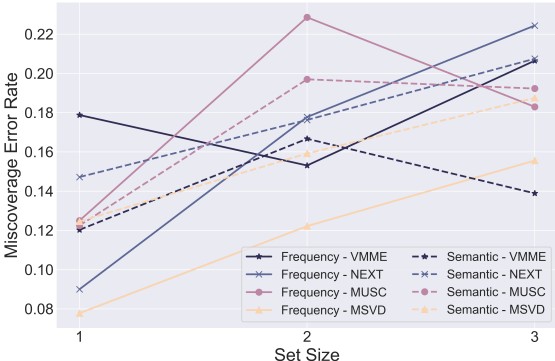

Figure 6: Stratified miscoverage rate at each size of prediction set on four VideoQA datasets utilizing the Gemini-1.5-Pro model. The solid line corresponds to Frequency, while the dashed line corresponds to Semantic Diversity.

MLLMs. The comparison results in Table 2 show that the employment of semantic diversity results in smaller APSSs across four MLLMs, indicating more efficient predictions, while the EER metric is always bounded by the risk level. This finding demonstrates the stability of our framework when using different reliability measures. It also points to a promising direction for further research on the impact of nonconformity score designs and their ability to represent agreement between question-and-answer pairs on prediction efficiency.

**Conditional Performance.**   As mentioned before, *TRON* bounds the correctness miscoverage rate marginally over the test set (i.e., on average). However, for some critical applications requiring more stringent guarantees, users demand that the error rate is bounded for a particular test data point. In other words, the risk level should be guaranteed for any subset of the data points. In the most general case, conditional coverage is impossible to achieve (Angelopoulos & Bates, 2021). Following prior work (Kumar et al., 2023; Su et al., 2024), we set $\alpha = \beta = 0.1$ (i.e., $\varepsilon = 0.19$) and evaluate the size-stratified miscoverage rate utilizing the Gemini-1.5-Pro model. As demonstrated in Figure 6, the miscoverage rate fluctuates up and down with the set size, with some values exceeding $\varepsilon$. It is worth noting that when we employ Semantic Diversity as the reliability measurement, TRON's conditional performance improves (e.g., the max miscoverage rate decreases from 0.2065 to 0.1667 on the VMME dataset). At this point, it's like a bag of tricks, and we can enhance the capability of reliability measurement to optimize the conditional performance of risk control.

## 5   CONCLUSION

Deploying MLLMs in practical VideoQA applications requires reliable risk control and assessment. In this paper, we present a general two-step framework for risk control by calibrating a conformal score to sample new generations and a nonconformity score to identify high-quality responses, which provides statistically and theoretically rigorous guarantees on the correctness miscoverage rate. Our method bridges the gap between SCP and closed-source MLLMs, and it can also be applied to other generative models (e.g., LLMs) that support sampling in various language generation tasks (see Appendix H). Furthermore, we conduct risk assessments and explore the issue of semantic redundancy in prediction sets within open-ended contexts for the first time, which results in a promising evaluation metric for MLLMs in open-ended VideoQA tasks via the average set size after deduplication. Additionally, we analyze reliability measurement in the development of the nonconformity score for more efficient predictions. We hope that our framework can be helpful for human-in-the-loop decision-making and human-AI teams.

LIMITATIONS

In our work, guarantees are marginal over the test set, not conditional to individual data points. In addition, our framework does not deviate from the fundamental exchangeability assumption of SCP, and we will explore risk control under distribution shift conditions in future work.

ACKNOWLEDGEMENTS

This work was supported by the National Natural Science Foundation of China (Grant No. 62122035) and the project of the Sichuan Provincial Department of Science and Technology (No. 2023YFQ0011). This research was completed during a visiting period at Southern University of Science and Technology.

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

## A  BACKGROUND OF SPLIT CONFORMAL PREDICTION

Split conformal prediction (SCP) is model-agnostic and distribution-free, and can offer statistically rigorous guarantees of ground-truth coverage on fresh test samples with minimal assumptions (e.g., data exchangeability) given a relatively small calibration set (Angelopoulos & Bates, 2021; Campos et al., 2024). SCP can transform any heuristic notion of uncertainty (i.e., the nonconformity score) from any model into a statistically rigorous one by constructing a prediction set adaptively to each particular input. Additionally, the size of each prediction set allows us to gain intuitive insights into the uncertainty of the current decision-making (i.e., the set becomes larger when the model is uncertain or the input is intrinsically hard) Ye et al. (2024); Kostumov et al. (2024).

Let's illustrate the framework utilizing classification problems. In the calibration set, $\{(X_i, Y_i^*)\}_{i=1}^N$, we define the nonconformity score (NS), $s_i$, to identify a heuristic notion of the uncertainty of the decision process of the model for the $i$-th calibration data. Here, we utilize one minus the softmax output corresponding to the true class to link the NS with the uncertainty condition of correctness closely. Then we calculate the $\frac{\lceil (n+1)(1-\alpha) \rceil}{n}$ (i.e., approximate $1 - \alpha$) quantile of all NSs (i.e., $\{s_1, ..., s_n\}$) in the calibration set and denote it as $\hat{q}$. Since there are approximately $1 - \alpha$ calibration samples whose uncertainty states corresponding to their true class satisfy (i.e., $\leq$) the threshold $\hat{q}$, we construct the prediction set of each test sample following $\mathcal{C}(X_{test}) = \{y \in [K] : s(X_{test}, y) \leq \hat{q}\}$. We expect that if the class $y$ satisfies $s(X_{test}, y) \leq \hat{q}$ (i.e., the uncertainty state of the current class $y$ satisfies the uncertainty condition corresponding to the true labels of approximate $1 - \alpha$ calibration samples), $y$ has the probability of $1 - \alpha$ to be the true label $Y_{test}^*$.

## B  PROOFS

### B.1  PROOF OF EQUATION 3

This section presents a standard proof of validity for the upper bound of the error rate, as defined in Eq.3. For completeness, we restate the definition of the conformal score in Eq.2. We define the minimum sampling threshold that ensures $Y_i \in \left\{ \hat{y}_m^{(i)} \right\}_{m=1}^{M_i}$ as

$$
\begin{aligned}
r_i &= r(X_i, Y_i) \\
&:= \sup \left\{ M_i : \forall M_i' < M_i, Y_i \notin \left\{ \hat{y}_m^{(i)} \right\}_{m=1}^{M_i'} \right\},
\end{aligned}
\tag{7}
$$

sort the conformal scores so that $r_1 \leq \cdots \leq r_N$, and calculate the $\frac{\lceil (N+1)(1-\alpha) \rceil}{N}$ empirical quantile of $r_1, \cdots, r_N$:

$$
\begin{aligned}
\hat{r} &:= \inf \left\{ r : \frac{|\{i : r_i \leq r\}|}{N} \geq \frac{\lceil (N+1)(1-\alpha) \rceil}{N} \right\} \\
&= r_{\lceil (N+1)(1-\alpha) \rceil}.
\end{aligned}
\tag{8}
$$

We proceed by noting the equality of the two events based on the definition in Eq.2:

$$
\left\{ y_{test} \notin \left\{ \hat{y}_m^{(test)} \right\}_{m=1}^{\hat{r}} \right\} = \left\{ r_{test} > \hat{r} \right\}.
\tag{9}
$$

By exchangeability[4] of the calibration and test data points, we have

$$
\mathbb{P}(r_{test} \leq r_i) = \frac{i}{N+1}.
\tag{10}
$$

---

[4]The standard framework of conformal prediction assumes exchangeability of all data points, and exchangeability is weaker than the independent and identically distributed (i.i.d.) assumption.

Then, we guarantee the upper bound of the probability that the candidate set fails to encompass an acceptable response based on a user-specified risk level, denoted as $\alpha$:

$$
\begin{aligned}
\mathbb{P}\left(y_{test} \notin \left\{\hat{y}_m^{(test)}\right\}_{m=1}^{\hat{r}}\right) &= \mathbb{P}\left(r_{test} > \hat{r}\right) \\
&= \mathbb{P}\left(r_{test} > r_{\lceil(N+1)(1-\alpha)\rceil}\right) \\
&= 1 - \mathbb{P}\left(r_{test} \leq r_{\lceil(N+1)(1-\alpha)\rceil}\right) \\
&= 1 - \frac{\lceil(N+1)(1-\alpha)\rceil}{N+1} \\
&\leq \alpha.
\end{aligned}
\tag{11}
$$

### B.2 Proof of equation 6

Given that we select trustworthy responses from each candidate set to build the prediction set, let us formally prove the upper bound of the error rate (i.e., $\varepsilon$) in Eq. 6, building upon the foundation established in Eq. 3. If there are no acceptable responses in the candidate set, event $\{y_{test} \notin \mathcal{C}\left(x_{test}\right)\}$ is a certain event, and the probability of miscoverage is

$$
\mathbb{P}\left(y_{test} \notin \mathcal{C}\left(x_{test}\right) \mid y_{test} \notin \left\{\hat{y}_m^{(test)}\right\}_{m=1}^{\hat{r}}\right) = \mathbb{P}\left(y_{test} \notin \left\{\hat{y}_m^{(test)}\right\}_{m=1}^{\hat{r}}\right).
\tag{12}
$$

Conversely, if at least one acceptable answer exists in the candidate set (i.e., $y_{test} \in \left\{\hat{y}_m^{(test)}\right\}_{m=1}^{\hat{r}}$), we then redefine the non-conformity score here. First, we determine the acceptable response of the $i$-th calibration data point $\hat{y}_{ref}^{(i)}$, which is semantically equivalent to the correct answer $Y_i$. Then we formulate the non-conformity score as

$$
\begin{aligned}
s_i &= s\left(X_i, Y_i\right) \\
&:= 1 - F\left(\hat{y}_{ref}^{(i)}\right),
\end{aligned}
\tag{13}
$$

where $F\left(\hat{y}_{ref}^{(i)}\right)$ is the frequency of $\hat{y}_{ref}^{(i)}$ in the candidate set. Similar to Eq. 8, we sort the non-conformal scores so that $s_1 \leq \cdots \leq s_N$, and define the $\frac{\lceil(N+1)(1-\beta)\rceil}{N}$ quantile of $s_1, \cdots, s_N$:

$$
\begin{aligned}
\hat{s} &:= \inf\left\{s : \frac{|\{i : s_i \leq s\}|}{N} \geq \frac{\lceil(N+1)(1-\beta)\rceil}{N}\right\} \\
&= s_{\lceil(N+1)(1-\beta)\rceil}.
\end{aligned}
\tag{14}
$$

Then, we construct the prediction set following

$$
\mathcal{C}\left(x_{test}\right) := \left\{\hat{y}_{m'}^{(test)} \in \left\{\hat{y}_m^{(test)}\right\}_{m=1}^{\hat{r}} : s\left(x_{test}, \hat{y}_{m'}^{(test)}\right) \leq \hat{s}\right\}.
\tag{15}
$$

Given $y_{test} \in \left\{\hat{y}_m^{(test)}\right\}_{m=1}^{\hat{r}}$, we obtain

$$
\begin{aligned}
\{y_{test} \notin \mathcal{C}\left(x_{test}\right)\} &= \{s\left(x_{test}, y_{test}\right) > \hat{s}\} \\
&= \{s_{test} > \hat{s}\}.
\end{aligned}
\tag{16}
$$

In this case, the probability of miscoverage is

$$
\mathbb{P}\left(y_{test} \notin \mathcal{C}\left(x_{test}\right) \mid y_{test} \in \left\{\hat{y}_m^{(test)}\right\}_{m=1}^{\hat{r}}\right) = \mathbb{P}\left(s_{test} > \hat{s} \mid y_{test} \notin \left\{\hat{y}_m^{(test)}\right\}_{m=1}^{\hat{r}}\right).
\tag{17}
$$

Similarity to Eq. 10, we have

$$
\begin{aligned}
\mathbb{P}\left(s_{test} > s_i\right) &= 1 - \mathbb{P}\left(s_{test} \leq s_i\right) \\
&= 1 - \frac{i}{N+1}.
\end{aligned}
\tag{18}
$$

Finally, we achieve risk control and guarantee the upper bound of the correctness miscoverage rate:

$$
\begin{aligned}
& \mathbb{P}\left\{y_{test} \notin \mathcal{C}\left(x_{test}\right)\right\} \\
& = \mathbb{P}\left(y_{test} \notin \left\{\hat{y}_m^{(test)}\right\}_{m=1}^{\hat{r}}\right) + \mathbb{P}\left(s_{test} > \hat{s} \mid y_{test} \in \left\{\hat{y}_m^{(test)}\right\}_{m=1}^{\hat{r}}\right) \\
& = 1 - \frac{\lceil (N+1)(1-\alpha) \rceil}{N+1} + \left(1 - \frac{\lceil (N+1)(1-\beta) \rceil}{N+1}\right) \cdot \frac{\lceil (N+1)(1-\alpha) \rceil}{N+1} \\
& = 1 - \frac{\lceil (N+1)(1-\beta) \rceil}{N+1} \cdot \frac{\lceil (N+1)(1-\alpha) \rceil}{N+1} \\
& \leq 1 - (1-\alpha)(1-\beta) \\
& \leq \alpha + \beta - \alpha\beta \\
& \leq \varepsilon.
\end{aligned}
\tag{19}
$$

## C  IMPLEMENTATION SPECIFICS

### C.1  SEMANTIC CLUSTERING PROCESS

---

**Algorithm 1:** The pseudo-code for semantic clustering.

---

**Input:** The candidate set of the $i$-th data point with $M_i$ response samples $\left\{\hat{y}_m^{(i)}\right\}_{m=1}^{M_i}$.

$\mathbb{C} = \{\}, \mathbb{F} = [\,]. \triangleright \mathbb{C}$ *is non-repeating*
**for** $m \leftarrow 1$ **to** $M_i$ **do**
$\quad \mathbb{L} = [m], f = 1.$
$\quad$ **for** $m' \leftarrow 1$ **to** $M_i$ **do**
$\quad\quad$ **if** $m' \neq m$ **then**
$\quad\quad\quad$ **if** $\hat{y}_{m'}^i \Leftrightarrow \hat{y}_m^i$ **then**
$\quad\quad\quad\quad \mathbb{L} \leftarrow \mathbb{L} + [m'];$
$\quad\quad\quad\quad f \leftarrow f + 1.$
$\quad \mathbb{C} \leftarrow \mathbb{C} + \{\mathbb{L}\};$
$\quad \mathbb{F} \leftarrow \mathbb{F} + [f].$
**Output:** Non-repeating semantic clusters $\mathbb{C}$, frequency of $M_i$ responses.

---

Note that since there may not be a semantically equivalent transfer between the sampled responses in open-ended tasks, the sum of the frequency scores corresponding to all responses in the candidate set may be uncertain, and we utilize the normalized frequency score for each response.

**Bidirectional Entailment.**  As noted in Section 3.2 and Algorithm 1, $A_1 \Leftrightarrow A_2$ represents that $A_1$ is semantically equivalent to $A_2$. Following prior studies (Duan et al., 2024; Lin et al., 2024; Farquhar et al., 2024), we utilize a Natural Language Inference (NLI) classifier, with the off-the-shelf DeBERTa-large-mnli model (He et al., 2021) as the backbone, for this evaluation. The NLI classifier typically takes $[A_1, A_2]$ as input and predicts logits for three distinct semantic relationships: entailment, neutral, and contradiction. Note that this function has directionality and we only consider $A_1$ and $A_2$ to be equivalent when the logit scores corresponding to the entailment are at their highest for both inputs $[Q \cup A_1, Q \cup A_2]$ and $[Q \cup A_2, Q \cup A_1]$ (i.e., bidirectional). Here, $Q$ denotes the question, and $\cup$ means we evaluate the semantic relationship between query-response pairs.

## D  DATASETS

Video-MME (VMME) (Fu et al., 2024) is a multi-modal evaluation benchmark for MLLMs in video analysis, which includes 900 manually selected videos totaling 254 hours and resulting in 2,700 MCQA pairs. NExT-QA (NEXT) (Xiao et al., 2021) is a benchmark for VideoQA that focuses on causal and temporal action reasoning, moving beyond simple scene descriptions. It features both multiple-choice and open-ended QA tasks. However, we only selected the multiple-choice

part for our experiments. We use the sub-test set of MCQA, which consists of 2,312 QA pairs. MUSIC-AVQA (MUSC) (Li et al., 2022a) is an open-ended VideoQA dataset designed to evaluate comprehensive multimodal understanding and spatio-temporal reasoning over audio-visual scenes. We use the test set of MUSIC-ACQA, which consists of 9,185 QA pairs. MSVD (Chen & Dolan, 2011) is an open-ended dataset that provides a large-scale, highly parallel text collection. We use 2,519 QA pairs from this dataset. In our experiments, we extracted the audio from the videos and integrated it with the corresponding visual and textual data before inputting them into the MLLMs.

## E  HYPERPARAMETERS

We employ multinominal sampling for responses which are used to construct prediction sets. The temperature of generation for all MLLMs is set to 1.0. All the experiments are conducted on a server with one Intel Xeon Gold 6326 CPU and 7 NVIDIA A6000 GPUs.

## F  EVALUATION METRICS

Empirical Error Rate (EER) evaluates the validity of our framework for risk control. In the first stage, we expect that, under the condition of exchangeability, we calibrate the number of response samples of each test data point based on the calibration set so that the candidate set encompasses at least one acceptable response with a specific probability. To this end, the EER on the test set, $\mathcal{D}_{test}$, should be bounded by the risk level $\alpha$ (i.e., $\leq \alpha$). At this point, we can approximate the closed-ended settings utilizing the candidate set. We redefine each sample in the test set as $(x_i, y_i)$ and the candidate set of the $i$-th data point as $\mathcal{S}(x_i)$. Then, EER is formulated as

$$\text{EER} = \frac{1}{|\mathcal{D}_{test}|} \sum_{(x_i, y_i) \in \mathcal{D}_{test}} \mathbb{1}\{y_i \in \mathcal{S}(x_i)\} \tag{20}$$

In the second stage, we expect that we can identify the acceptable response within the candidate set, which is also said the prediction set encompasses the correct answer, based on the nonconformity score with a user-specified probability. To this end, the EER on the test set should be bound by $\alpha$ and another risk level $\beta$ (i.e., $\leq \alpha + \beta - \alpha\beta$.). In the evaluations, we set $\alpha = 0.1$ by default and measure EER at various values of $\beta$. At this point, the upper bound of EER is $0.1 + \beta - 0.1\beta$, and EER is formulated as

$$\text{EER} = \frac{1}{|\mathcal{D}_{test}|} \sum_{(x_i, y_i) \in \mathcal{D}_{test}} \mathbb{1}\{y_i \in \mathcal{C}(x_i)\} \tag{21}$$

Average Prediction Set Size (APSS) measures the average size of all prediction sets required to guarantee the EER on the test set. A larger APSS indicates greater uncertainty/risk, while a smaller APSS reflects higher efficiency and confidence. At this point, APSS is formulated as

$$\text{APSS} = \frac{1}{|\mathcal{D}_{test}|} \sum_{(x_i, y_i) \in \mathcal{D}_{test}} |\mathcal{C}(x_i)| \tag{22}$$

Accuracy (ACC) assesses the correctness of MLLMs' predictions, serving as a foundational benchmark for model performance. In our experiments, we employ the most frequent (reliable) response within the candidate set of each test data point, denoted as $\hat{y}_{ref}^{(i)}$, as the model output to measure the total accuracy, which is formulated as.

$$\text{ACC} = \frac{1}{|\mathcal{D}_{test}|} \sum_{(x_i, y_i) \in \mathcal{D}_{test}} \mathbb{1}\left\{\hat{y}_{ref}^{(i)} = y_i\right\} \tag{23}$$

## G    ADDITIONAL EXPERIMENTAL ANALYSIS

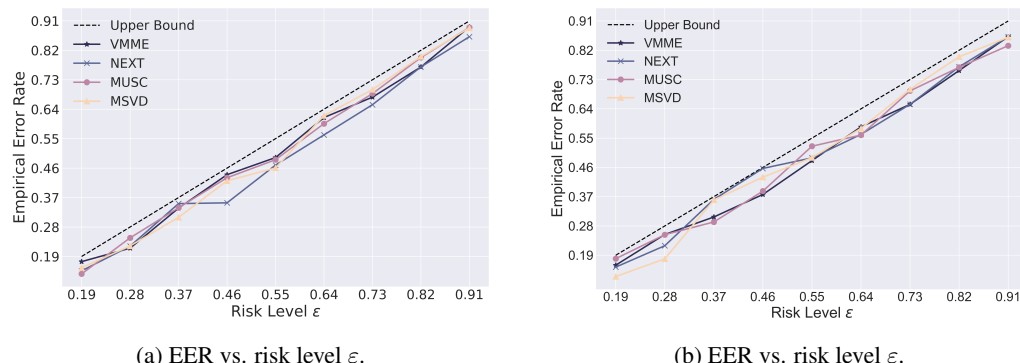

(a) EER vs. risk level $\varepsilon$.               (b) EER vs. risk level $\varepsilon$.

Figure 7: (a) Gemini-1.5-Flash; (b) Gemini-1.5-Pro. We fix the risk level in the first stage at 0.1 (i.e., $\alpha = 0.1$) and evaluate the correctness miscoverage rate at various values of $\beta$.

## H    GENERALIZABILITY OF *TRON*

As mentioned before, *TRON* applies to any generative language models that support sampling (or repeatedly responding to the same query) on various open-ended tasks. To evaluate its generalizability, we employ *TRON* on the (1) open-book conversational QA dataset, CoQA (Reddy et al., 2019), and the (2) closed-book reading comprehension dataset, TriviaQA (Joshi et al., 2017), utilizing LLMs, LLaMA-3-70B-Instruct (AI@Meta, 2024) and LLaMA-3.1-70B-Instruct. Note that we use LangChain with DeepInfra for LLaMA-3-70B-Instruct [5] and LLaMA-3.1-70B-Instruct [6]. We randomly select 2,000 questions from the development split of CoQA and 2,000 questions from the training split of TriviaQA, and set the split ratio between the calibration and test set to 0.5 for both datasets. Empirical results of the correctness miscoverage rate, obtained from 5 trials with randomly allocated calibration and test data at various user-specified risk levels are shown in Figure 8.

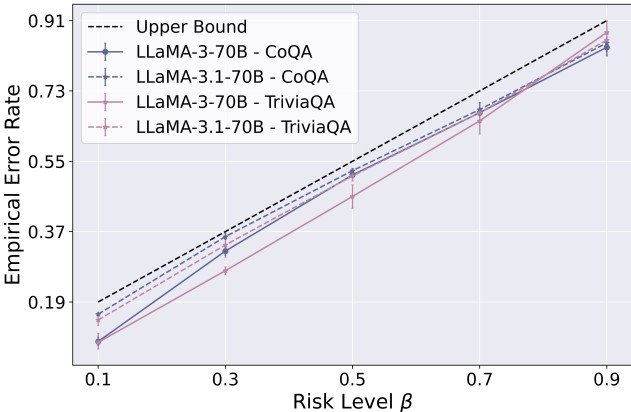

Figure 8: Results of the EER metric at various risk levels on the CoQA and TriviaQA tasks utilizing LLaMA-3-70B-Instruct and LLaMA-3.1-70B-Instruct. Note that we fix the risk level $\alpha$ to 0.1.

We also utilize the PandaGPT-13B model and GPT-4o [7] on the Visual Question Answering (VQA) dataset (Antol et al., 2015) for image understanding tasks. We randomly select 1,200 VQA samples and employ 200 samples as calibration data and 1,000 samples as test data. We set $\alpha$ to 0.1 and evaluate the ERR metric at various user-specified risk levels, $\beta$, as shown in Figure 9.

---

[5]https://deepinfra.com/meta-llama/Meta-Llama-3-70B-Instruct

[6]https://deepinfra.com/meta-llama/Meta-Llama-3.1-70B-Instruct

[7]https://platform.openai.com/docs/models/gpt-4o

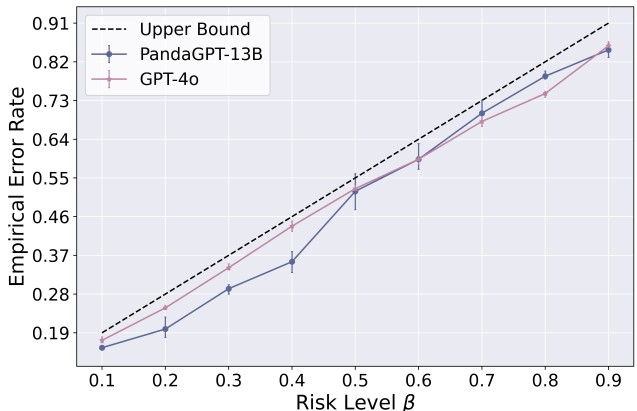

Figure 9: Results of the EER metric, obtained from 5 trials with randomly allocated calibration and test data at various risk levels on the VQA dataset utilizing PandaGPT-13B and GPT-4o.

Empirical results demonstrate that *TRON* can also provide risk management for both MLLMs and LLMs on image understanding, conversational QA, and reading comprehension tasks. Note that while the theoretical guarantee of correctness coverage is rigorous, there can be minor fluctuations of empirical miscoverage rates that exceed the user-specified risk levels in practice due to finite-sample variability (Angelopoulos & Bates, 2021; Ye et al., 2024).

**Comparisons on MCQA tasks.** We also compare *TRON* with the existing method for LLMs on MCQA tasks, termed Least Ambiguous set-valued Classifiers (*LAC*) (Sadinle et al., 2019; Ye et al., 2024), which has been proven to produce prediction sets with the smallest average size (Sadinle et al., 2019). *LAC* defines the nonconformal score function as one minus the softmax score corresponding to the true label. We adopt MMLU (Hendrycks et al., 2021) as the evaluation dataset utilizing LLaMA-3-8B-Instruct [8] and LLaMA-3.1-8B-Instruct [9]. To maintain consistency with previous work settings (Ye et al., 2024), we only select samples that cover at least one correct response within the candidate set when $M$ is set to 5 (i.e., $\alpha = 0$). We set the splitting ratio between calibration and test set to 0.5 and $\beta$ to 0.1, and evaluate the APSS metric. Table 3 demonstrates that when sampling 10 times utilizing the LLaMA-3-8B-Instruct model, *TRON*'s performance can already match that of *LAC*, and when $M$ is 20, *TRON* is more predictive efficient than *LAC*. Additionally, when the sampling size is set to 10, *TRON* outperforms *LAC*. Our insight is that since model internal logits can induce miscalibrated biases due to arbitrarily overfit or otherwise untrustworthy issues, we analyze the output distribution of the model through sampling, which is more sound.

| Model | $LAC$ | $TRON$ ($M = 5$) | $TRON$ ($M = 10$) | $TRON$ ($M = 20$) |
|---|---|---|---|---|
| LLaMA-3-8B-Instruct | $2.93_{(2)}$ | $3.06_{(0)}$ | $2.93_{(7)}$ | $\mathbf{2.76_{(3)}}$ |
| LLaMA-3.1-8B-Instruct | $2.57_{(8)}$ | $2.61_{(3)}$ | $2.53_{(6)}$ | $\mathbf{2.50_{(4)}}$ |

Table 3: The APSS metric ($\downarrow$) on the MMLU dataset.

---

[8]https://huggingface.co/meta-llama/Meta-Llama-3-8B-Instruct
[9]https://huggingface.co/meta-llama/Meta-Llama-3.1-8B-Instruct

