# OpenReview forum: "Sample then Identify: A General Framework for Risk Control and Assessment in Multimodal Large Language Models"
_ICLR.cc/2025/Conference — ICLR 2025 Spotlight_

### Official Review · Reviewer_pRzK · 2024-10-31

**Soundness:** 3
**Presentation:** 2
**Contribution:** 3
**Rating:** 8
**Confidence:** 4

**Summary:**

The paper presents **TRON**, a two-step framework designed for **risk control and assessment** in multimodal large language models (MLLMs), particularly for **Video Question Answering (VideoQA)** tasks. Addressing challenges in dynamic and open-ended environments, TRON leverages **Split Conformal Prediction (SCP)**, introducing a **conformal score** for sampling response sets and a **nonconformity score** for identifying high-quality responses. Through experiments on several VideoQA datasets, TRON demonstrates the ability to achieve desired error rates across various **user-specified risk levels**. It is also noted for exploring the concept of **semantic redundancy** in prediction sets as an evaluation metric, an area not previously investigated in open-ended contexts.

**Strengths:**

1. **Innovative Framework**: The introduction of TRON, a two-step risk control and assessment framework, contributes significantly to the field of MLLM evaluation in both open-ended and closed-ended VideoQA tasks. Its flexibility in applying conformal prediction in open-ended contexts is commendable.

2. **Novel Conformal and Nonconformity Scores**: The paper proposes a unique conformal score for setting the minimum sample size in open-ended tasks and a nonconformity score based on self-consistency theory. These scores provide a rigorous approach to risk control.

3. **Addressing Uncertainty with Redundancy Analysis**: The evaluation of **semantic redundancy** in open-ended settings introduces a new angle to uncertainty measurement, providing a promising metric that complements traditional accuracy.

4. **Comprehensive Experimental Evaluation**: The experiments span multiple datasets, risk levels, and different types of MLLMs, offering a thorough assessment of TRON’s effectiveness in diverse conditions.

**Weaknesses:**

1. **Limited Discussion of Practicality and Adaptability**: While TRON provides theoretical guarantees, the practical aspects, such as computational overhead and applicability in real-world scenarios, could be discussed in greater depth.

2. **Insufficient Baseline Comparisons**: The paper lacks a comparison with other standard risk control methods or frameworks that may be relevant, particularly in closed-ended settings or previous SCP applications in MLLMs.

3. **Complexity in Method Presentation**: Some sections of the methodology, particularly the derivation of the conformal and nonconformity scores, lack clarity, which could challenge readers unfamiliar with SCP.

4. **Inconsistent Evaluation Details**: Although the experiments are extensive, details on some evaluation metrics, like APSS and their implications for different risk levels, could be better explained. The choice of models and how each metric was applied in open-ended versus closed-ended settings was also not consistently clarified.

**Questions:**

1. **What is the expected computational impact** of using TRON in large-scale applications or real-time risk assessment tasks? Could you provide a more detailed explanation or metrics regarding the processing time required for each step?

2. **Baseline Method Comparison**: Have you considered comparing TRON with simpler risk control baselines? For instance, would a heuristic-based risk control method suffice for certain types of tasks in closed-ended VideoQA? If not, could you clarify how TRON performs specifically better in such cases?

3. **Semantic Redundancy in Open-Ended Tasks**: How does the semantic redundancy analysis handle responses that may be lexically distinct but only partially semantically equivalent? Could this approach potentially overlook responses with subtle but important semantic differences?

4. **Alternative Conformal and Nonconformity Scoring Methods**: Could the proposed conformal and nonconformity scores be further enhanced by alternative methods, such as clustering-based or confidence-based approaches beyond self-consistency theory? If so, what would be the implications for TRON’s current framework?

5. **Additional Real-World Validation**: Have there been any attempts to validate TRON in real-world or industry-specific VideoQA tasks, possibly in collaboration with industry partners? If so, could you share any preliminary insights on its practical performance and potential adaptations? If not, do you plan to include such validation in future work?

---

> ### Author Response · Authors · 2024-11-16
> **Responses to Reviewer pRzK Comments on Four Weaknesses**
>
> > **Response to weakness 1:**
>
> Thanks for your valuable suggestions. We have added empirical evaluations for TRON in the tasks of conversational question answering, reading comprehension, and image understanding in **Appendix H (Figures 8 and 9)**. Results demonstrate that TRON is applicable to generative language models (e.g., VLMs and LLMs) on various open-ended tasks, which is primarily attributed to TRON establishing statistical criteria by analyzing the output distribution of the model.
>
> Additionally, given that practical applications place more emphasis on the **conditional performance of risk control**, we analyze how close our method comes to approximating condition coverage utilizing the size-stratified miscoverage metric in **Section 4.4 (Figure 6)**. When we employ Semantic Diversity as the reliability measurement, TRON’s conditional performance improves (e.g., the max miscoverage rate decreases from 0.2065 to **0.1667** on the VMME dataset when the risk level is set to 0.19).
>
> > **Response to weakness 2:**
>
> Thanks for your feedback. Since we approximate the candidate set, in open-ended language generation tasks, as multiple-choice options at a user-specified risk level for the first time, we compare TRON with Least Ambiguous set-valued Classifiers (LAC), which has been proven to produce prediction sets with the smallest average size in **closed-ended settings**. Empirical results of the APSS metric ($\downarrow$) are shown in **Table 3 in Appendix H**. TRON significantly outperforms LAC, and increasing the sampling size within the allowed sampling cost will make TRON more prediction-efficient.
>
> | Model\Method          | LAC    | TRON (M = 5) | TRON (M = 10) | TRON (M = 20) |
> |------------------|--------|--------------|---------------|---------------|
> | LLaMA-3-8B-Instruct        | $2.93_2$ | $3.06_0$ | $2.93_7$| $2.76_3$|
> | LLaMA-3.1-8B-Instruct        | $2.57_8$ | $2.61_3$ | $2.53_6$| $2.50_4$|
>
> > **Response to weakness 3:**
>
> Thanks for your suggestions for improving the paper. We have illustrated the base framework of conformal prediction utilizing classification problems in **Appendix A**, and provided a detailed derivation of the risk management at both the sampling and identifying processes in **Appendix B**.
>
> > **Response to weakness 4:**
>
> Thanks for your suggestion. We have provided a detailed explanation of the meanings and functions of each metric in Appendix F. APSS refers to the average size of the prediction sets for all test data on the test set, used to assist accuracy in evaluating the model's uncertainty on the test set. APSS decreases as the risk level increases, meaning that the allowable error rate is raised, as shown in Figure 3. We generally fix the risk level and use APSS to evaluate the uncertainty of different models on the same test set to assess their performance. Furthermore, APSS provides a consistent evaluation of uncertainty in both open-ended and closed-ended tasks. However, in open-ended tasks, we observe that there is semantic redundancy in the prediction sets. Therefore, we analyze the impact of semantic redundancy on APSS and the final risk assessment in Section 4.3.

---

> ### Author Response · Authors · 2024-11-16
> **Responses to Reviewer pRzK Comments on Five Questions**
>
> > **Response to question 1:**
>
> In practical applications where calibration data has already been set up, TRON requires obtaining M responses from the model output space based on the minimum number of samples derived from the calibration data. This process has a lower computational cost compared to beam search because we use multinomial sampling.
> The main computational cost lies in evaluating the reliability of each response, which depends on the risk requirements of the practical application. In high-risk applications, such as medical diagnosis, it is clearly insufficient to evaluate each response solely based on frequency. Additionally, if there is a distribution shift problem, meaning the exchangeability condition is not met, we need to deploy more powerful models to mitigate the issue of uneven distribution, as we primarily rely on analyzing the model's output distribution. If we can access the model's internal logit information, we can also increase the computational cost to calculate the entropy of each response, thereby improving TRON's prediction efficiency.
>
> > **Response to question 2:**
>
> We have added a comparison of the prediction efficiency of TRON and Least Ambiguous set-valued Classifiers (LAC), which has been proven to produce prediction sets with the smallest average size in closed-ended settings in **Appendix H (Table 3)**. Empirical results demonstrate that TRON produces more efficient predictions. Furthermore, TRON is similar to **a bag of tricks**. As mentioned in the last paragraph of Section 3.2, the reliability function $F$ can be any measurement that reflects the trustworthiness of each response. TRON provides, for the first time, a guarantee for the miscoverage rate in language generation tasks in open-ended settings. Within this broader framework, we can individually optimize each step to enhance the overall risk management capability.
>
> > **Response to question 3:**
>
> ***Firstly***, following prior work [1][2], we utilize a Natural Language Inference (NLI) classifier with DeBERTa-large-mnli as the backbone to evaluate the semantic equivalence between two responses. As mentioned in Appendix C, If two responses are predicted to have a bidirectional entailment relationship, we consider that semantic redundancy has occurred. ***Secondly***,  the approach may potentially overlook responses with subtle but important semantic differences, because there is no perfect method to evaluate the semantic relationship between two responses. At this point, we can combine methods such as the ROUGE-L score for multiple checks to achieve a more rigorous assessment of semantic equivalence, but this would impose an additional computational cost.
>
>
> > **Response to question 4:**
>
> Yes, as mentioned in Section 4.4, by incorporating semantic similarity information and using Semantic Diversity as the reliability measurement, both the prediction efficiency and conditional performance of TRON are improved. Our motivation of the reference to self-consistency is that when constrained by a black-box setting, we can define the nonconformity score solely by analyzing the model's output distribution. A more accurate reliability measurement will enhance the performance of TRON.
>
> > **Response to question 5:**
>
> TRON can be combined with conformal risk control [3] and prompt risk control [4] to achieve task-specific performance control in various language generation tasks such as VideoQA and Vision-Language QA. Taking the task of diagnosing Parkinson's disease in medical imaging as an example, we can prompt a vision-language model (VLM) to identify the lesion areas in brain imaging slices. At this point, we can control the precision of the VLM's identification of lesion areas by defining the nonconformity score as the false discovery rate for each sample.
>
> ---
>
> ### References
>
> [1] Semantic uncertainty: Linguistic invariances for uncertainty estimation in natural language generation (ICLR 2023).
>
> [2] Detecting hallucinations in large language models using semantic entropy (Nature 2024).
>
> [3] Conformal Risk Control (ICLR 2024)
>
> [4] Prompt Risk Control: A Rigorous Framework for Responsible Deployment of Large Language Models (ICLR 2024)

---

> ### Author Response · Authors · 2024-11-21
> **Rebuttal Revision**
>
> We would like to thank you for your detailed and considerate reviews of our paper. We have uploaded a modified version of our paper that incorporates your comments.
>
> - In the main text, we add a conditional performance analysis of TRON utilizing the size-stratified miscoverage rate in Section 4.4, Paragraph Conditional Miscoverage Rate (Figure 6) for the applicability in real-world critical applications requiring more stringent guarantees. Empirical results demonstrate that the miscoverage rate varies at different set sizes, and the conditional performance of TRON significantly improves by integrating the semantic similarity information into the reliability measurement (e.g., the maximum miscoverage rate decreases from 0.2065 to **0.1667** on the VMME dataset when the upper bound is set to 0.19). ***(Corresponding to Weakness 1)***
>
> - In Appendix A, we add a brief illustration to conformal prediction utilizing classification tasks. ***(Corresponding to Weakness 3)***
>
> - In Appendix H, we evaluate the generalizability of TRON to other generative language models in various open-ended tasks. We consider the **(1) open-book conversational QA** dataset, CoQA, and the **(2) closed-book reading comprehension** dataset, TriviaQA, utilizing the large language models (LLMs), LLaMA-3-70B-Instruct and LLaMA-3.1-70B-Instruct (from LangChain DeepInfra). Additionally, we employ PandaGPT-13B and GPT-4o on the Vision Question Answering (VQA) dataset for **(3) image understanding** tasks. Empirical results demonstrate that TRON can also provide risk management for both MLLMs and LLMs on image understanding, conversational QA, and reading comprehension tasks (Figures 8 and 9). ***(Corresponding to Weakness 1)***
>
> - Furthermore, we compare TRON with Least Ambiguous set-valued Classifiers (LAC) in closed-ended settings, which has been proven to produce prediction sets with the smallest average size. Evaluations on the MMLU dataset (MCQA) utilizing both LLaMA-3-8B-Instruct and LLaMA-3.1-8B-Instruct demonstrate that TRON is more predictive efficient than LAC (Table 3). ***(Corresponding to Weakness 2)***
>
> Overall, we hope that these changes have addressed your concerns. We would be grateful for the opportunity to engage further with you to discuss any remaining questions or concerns you may have.

---

> ### Author Response · Authors · 2024-11-23
> **Supplementary Response to Question 5**
>
> > Additional Real-World Validation: Have there been any attempts to validate TRON in real-world or industry-specific VideoQA tasks, possibly in collaboration with industry partners? If so, could you share any preliminary insights on its practical performance and potential adaptations? If not, do you plan to include such validation in future work?
>
> Let's take a medical video analysis task as an example. Given $N$ medical video-report pairs $\lbrace(X_i,Y_i^*) \rbrace_{i=1}^N$ and a new test data $(X_t,Y_t^*)$ (or $(X_{test},Y_{test}^*)$)
>
> ***Step 1.*** For each medical query based on the video, we sample $m$ generations from the model, denoted as $\mathcal{C_m} (X_i)=\lbrace  \hat Y_j^{(i)}\rbrace_{j=1}^m$. Then, we define the loss of miscoverage by the candidate set as $\mathcal{l}(\mathcal{C_m}(X_i),Y_i^*)= \mathcal{1}\lbrace Y_i^*∉ \mathcal{C_m}(X_i)\rbrace$, and the loss is non-increasing in $m$.
>
> We set the size of the candidate set to $X_{test}$ to be $\hat m= inf \lbrace m: \frac{A_N (m) +1}{N+1}\leqslant \alpha \rbrace$ $=inf\lbrace m: A_N(m) \leqslant \alpha (N+1)-1\rbrace$, where $A_N(m)=\sum_\limits{i=1}^{N} l(\mathcal{C}_{m}(X_i),Y_i^*).$ Since $A_N(m)$ is monotone in $m$, we can efficiently search for $\hat m$ by binary search to arbitrary precision.
>
> Given that $l (\mathcal{C}_{\hat m} (X_t),Y_t^*) \leq 1(\in \lbrace 0,1\rbrace)$, we obtain
>
> $A_{N+1}(\hat m)=\sum\limits_{i=1}^{N+1}l(\mathcal{C}_{\hat m}(X_i),Y_i^*)$
>
> $= A_N(\hat m)+l(\mathcal(C)_{\hat m}(X_t),Y_t^*)$
>
> $\leq A_N(\hat m)+1$
>
> $\leq \alpha (N+1)$
>
> By the exchangeability of $N$ calibration data points and the test data point, we have $l(\mathcal(C_{\hat m}(X_t),Y_t^*)$~$Uniform(\lbrace l(\mathcal{C_{\hat m}}(X_1),Y_1^*),...,l(\mathcal{C}_{\hat m}(X_t),Y_t^*)\rbrace)$. Then, we guarantee the error rate of the candidate set of the test data point failing to encompass an acceptable generation
>
> $\mathbb{E}\[ l \(\mathcal{C_{\hat m}} \(X_t\),Y_t^*\)\]= \frac{1}{N+1}  \sum\limits_{i=1}^{N+1}l(\mathcal{C}_{\hat m}(X_i),Y_i^*)$
>
> $=\frac{A_{N+1}({\hat m})}{N+1}$
>
> $\leq \alpha$
>
> ***Step 2.*** We assume these $N+1$ data points have at least one acceptable generation within their individual candidate set (i.e., $\alpha=0$). We then post-process the candidate set of the test data point by selecting high-quality responses to construct a prediction set, following
> $\mathcal{P}_{\hat t}(X_t)=\lbrace {\hat Y_k^{(test)}}\in \lbrace{\hat Y_j^{(test)}}\rbrace_j^{\hat m} : F({\hat Y_k^{(test)}})\geq 1-{\hat t}\rbrace$,
> where $F$(·)  measures the reliability of each generation within the candidate set and $\hat t$ is defined as
> $\hat t = inf\lbrace t:\frac{R_N(t)+1}{N+1}\leq\delta\rbrace$
>
> , where $R_N(t) = \sum\limits_{i=1}^N r(\mathcal{P}_t(X_i),Y_i^*)=\sum\limits_i^N 1-  \frac{\mathcal{P}_t(X_i) \cap Y_i^*}{|\mathcal{P}_t(X_i)|}$.
>
> Similarly, we control the **false discovery rate (FDR)** of a new medical report $\mathbb{E}[r(\mathcal{P}_{\hat t}(X_t),Y_t^*)] \leq \delta$
>
> At this point, we can sample multiple medical video reports and extract reliable sub-claims using $F$ to form a prediction set (i.e., a new medical report), while controlling the FDR within the prediction set. This is an adapted framework of TRON in real-world applications. We hope that our responses have addressed your concerns, and we would greatly appreciate it if you could reconsider the score. Thank you again for your thoughtful review.

---

> ### Author Response · Authors · 2024-11-25
> **Have we addressed your concern?**
>
> Dear Reviewer pRzK,
>
> Thank you again for taking the time to review our paper and providing detailed feedback. As the end of the discussion period is approaching, we want to follow up to see if you have any additional questions we have not addressed, or if there is anything else we can help clarify. We have tried responding to the comments from your initial review, and we are more than happy to discuss any points further. Thank you!
>
> Authors of paper 4201

---

> > ### Comment · Reviewer_pRzK · 2024-11-25
> > **Reply to Authors' responses**
> >
> > Thank you to the authors for their efforts and for addressing my concerns. I have considered this paper acceptable from the beginning and appreciate your responses to my questions. However, the scope and impact of the innovations presented in this work might be somewhat limited within the field. Therefore, I will retain my current score, but I am confident that the authors will produce even more impactful work in the future that deserves the highest recognition.

---

> > > ### Author Response · Authors · 2024-11-25
> > > **Appreciation for Recognition**
> > >
> > > Thank you for your recognition and encouragement. We appreciate your valuable insights and will strive to make a broader impact in our future work. Thanks again for taking the time to review our responses and revision.

---

> > > > ### Comment · Reviewer_pRzK · 2024-11-30
> > > > **Reply to Authors' responses**
> > > >
> > > > I referred to the author's final revised version and implemented some experiments using your ideas. With your responses, I now have a more comprehensive understanding of the work. I hope you will continue your efforts. I have increased my score.

---

### Official Review · Reviewer_Tw6D · 2024-11-03

**Soundness:** 3
**Presentation:** 3
**Contribution:** 2
**Rating:** 6
**Confidence:** 4

**Summary:**

The paper deals with risk control and assessment for MLLMs. To address the issues of existing work relying on the internal model logits and working in the multiple-choice setting, the authors propose a TRON, a two-step framework for MLLMs supporting sampling for both open-ended and close-ended scenarios. TRON allows controlling error rates by sampling response sets of minimum size and identifying high-quality responses using self-consistency theory. The experiments on VideoQA demonstrate the efficacy on eight MLLMs.

**Strengths:**

- The paper is easy to follow and understand.
- The proposed method extends SCP to open-ended scenarios by estimating the confidence from frequency.

**Weaknesses:**

- The confidence estimation in Step 2 relies on the prediction of another model(e.g. DeBERTa-large-mnli). Then it should be at least discussed on the reliability as the semantic classifier. Otherwise, it makes the identification of risk control less convincing.

- It is unclear how the silence percentage is conducted on the audio, and how the conclusion ‘introduce audio modality enhances the confidence level’ (Line371-372) is made. It is shown in Fig. 4 that increasing SPs leads to higher APSS. Also, introducing audio modality seems to only improve VideoLLaMA with SP <50%.

- It seems that the proposed method is general to LLMs as well. How does the proposed method work for LLMs for both open-ended and close-ended scenarios? The paper claims this advantage but shows no experimental results. This would make the paper more impactful.

- How is the proposed method compared with existing methods for LLMs on such as MCQA? Such a comparison would make the paper more comprehensive.

**Questions:**

- It seems that the best practice of the ratio of the calibration and test set is model-dependent. Is there any insight on the ratio selection when applied to different MLLMs?

---

> ### Author Response · Authors · 2024-11-16
> **Responses to Reviewer Tw6D Comments on Four Weaknesses**
>
> > **Response to weakness 1:**
>
> Thank you for your valuable feedback. **Firstly**, we have provided detailed explanations of bidirectional entailment in Appendix C. The Natural Language Inference (NLI) classifier with DeBERTa-large-mnli as the backbone has been widely recognized and employed to evaluate the semantic equivalence [1][2][3][4]. **Secondly**, we measure the reliability of each response based on the number of responses that are semantically equivalent to it, and utilize the same evaluation method for both calibration and test data points. Since the nonconformity score is strictly linked with the correct response, the final guarantees of miscoverage will not be impacted. **Furthermore**, errors exist for each data point because it is impossible to find a perfect semantic equivalence evaluation method. However, this does not affect risk control under the condition of exchangeability, as the criteria are consistent for each data point. This is also the core point of why conformal prediction can provide statistically rigorous guarantees. We are eliminating anomalies under consistent criteria to achieve risk control.
>
> > **Response to weakness 2:**
>
> Thank you for pointing out it. **Firstly**, Silence Percentage (SP) refers to the proportion of randomly muted audio segments in a video **(i.e., larger SP, less audio modality information)**. For each video, varying levels of muting are applied to create different SP levels. **Secondly**, the average prediction set size (APSS) reflects the uncertainty of model decision-making[5][6]. The introduction of the audio modality generally enhances the model's confidence by providing complementary information to the visual and text modality. As shown in Figure 4, the APSS metric significantly decreases (i.e., low uncertainty) as SP decreases from 100% to 0% (i.e., gradually incorporating the audio modality). **Additionally**, increasing SP means that the audio modality information decreases, leading to an increase in model uncertainty, which results in a larger APSS. **Furthermore**, our motivation is to complement the empirical findings of the two studies [5][6]. The study [5] utilizes APSS to evaluate the uncertainty of large language models (LLMs) and the study [6] evaluates the uncertainty of visual-language models (VLMs). Neither of them considered audio modality information. Since the accuracy metric does not provide a comprehensive evaluation of the model's performance (e.g., when SP $\leq$ 50%, as SP increases, uncertainty increases while the accuracy metric also rises, utilizing the VideoLLaMA-7B model) [5][6], we use APSS to assess the uncertainty of MLLMs before and after introducing the audio modality. From Figure 4, we observe that as the proportion of audio modality information is gradually removed (i.e., SP ↑), the model's uncertainty increases and accuracy decreases generally.
>
> > **Response to weakness 3:**
>
> Thanks for your valuable suggestions to improve the paper. TRON is applicable to generative language models on various open-ended tasks, as it formulates the criterion by analyzing the output distribution of the generative model. In the rebuttal revision, we have evaluated TRON on the **open-book conversational QA** dataset, CoQA [1], and the **closed-book reading comprehension** dataset, TriviaQA [2], utilizing the large language model, LLaMA-3-70B-Instruct. Additionally, we employ PandaGPT-13B on the Visual Question Answering (VQA) dataset [3] for **image understanding** tasks. Empirical results have been added to **Appendix H (Figure 8 and 9)**.
>
> > **Response to weakness 4:**
>
> Thanks for your thoughtful insights. We have compared TRON with Least Ambiguous set-valued Classifiers (LAC), which has been proven to produce prediction sets with the smallest average size, on the MMLU dataset utilizing LLaMA-3-8B-Instruct and LLaMA-3.1-8B-Instruct. Empirical results of the APSS metric ($\downarrow$) are shown in **Table 3 in Appendix H**.
>
> | Model\Method          | LAC    | TRON (M = 5) | TRON (M = 10) | TRON (M = 20) |
> |------------------|--------|--------------|---------------|---------------|
> | LLaMA-3-8B-Instruct        | $2.93_2$ | $3.06_0$ | $2.93_7$| $2.76_3$|
> | LLaMA-3.1-8B-Instruct        | $2.57_8$ | $2.61_3$ | $2.53_6$| $2.50_4$|
> ---
>
> ### References
>
> [1] Semantic uncertainty: Linguistic invariances for uncertainty estimation in natural language generation (ICLR 2023).
>
> [2] Shifting Attention to Relevance: Towards the Predictive Uncertainty Quantification of Free-Form Large Language Models (ACL 2024).
>
> [3] Generating with Confidence: Uncertainty Quantification for Black-box Large Language Models (TMLR 2024).
>
> [4] Detecting hallucinations in large language models using semantic entropy (Nature 2024).
>
> [5] Benchmarking LLMs via Uncertainty Quantification (NeurIPS 2024).
>
> [6] Uncertainty-Aware Evaluation for Vision-Language Models.

---

> ### Author Response · Authors · 2024-11-16
> **Responses to Reviewer Tw6D Comments on One Question**
>
> > Question: It seems that the best practice of the ratio of the calibration and test set is model-dependent. Is there any insight on the ratio selection when applied to different MLLMs?
>
>
> **Response:** Conformal prediction relies on the exchangeability of calibration and test data. When the data distribution of the calibration set cannot encompass the test data, a distribution shift problem arises. In this case, we should expand the range of the calibration set.
>
> In our view, the distribution shift is determined by the generative language model. For example, when the model performs very well on both the calibration and test sets, the output distributions (i.e., statistical metrics that define the nonconformity score) on the calibration and test sets will be similar, allowing us to choose a smaller calibration set ratio. However, if the model struggles with the test data, we should increase the size of the calibration set as much as possible. Theoretically, this means enhancing the generalization ability of the nonconformity scores on the calibration set to the test data.

---

> ### Author Response · Authors · 2024-11-21
> **Rebuttal Revision**
>
> We would like to thank you for your detailed and considerate reviews of our paper. We have uploaded a modified version of our paper that incorporates your comments.
>
> - In the main text, we add a conditional performance analysis of TRON utilizing the size-stratified miscoverage rate in Section 4.4, Paragraph Conditional Miscoverage Rate (Figure 6) for the applicability in real-world critical applications requiring more stringent guarantees. Empirical results demonstrate that the miscoverage rate varies at different set sizes, and the conditional performance of TRON significantly improves by integrating the semantic similarity information into the reliability measurement (e.g., the maximum miscoverage rate decreases from 0.2065 to **0.1667** on the VMME dataset when the upper bound is set to 0.19).
>
> - In Appendix A, we add a brief illustration to conformal prediction utilizing classification tasks.
>
> - In Appendix H, we evaluate the generalizability of TRON to other generative language models in various open-ended tasks. We consider the **(1) open-book conversational QA** dataset, CoQA, and the **(2) closed-book reading comprehension** dataset, TriviaQA, utilizing the large language models (LLMs), LLaMA-3-70B-Instruct and LLaMA-3.1-70B-Instruct (from LangChain DeepInfra). Additionally, we employ PandaGPT-13B and GPT-4o on the Vision Question Answering (VQA) dataset for **(3) image understanding** tasks. Empirical results demonstrate that TRON can also provide risk management for both MLLMs and LLMs on image understanding, conversational QA, and reading comprehension tasks (Figures 8 and 9). ***(Corresponding to Weakness 3)***
>
> - Furthermore, we compare TRON with Least Ambiguous set-valued Classifiers (LAC) in closed-ended settings, which has been proven to produce prediction sets with the smallest average size. Evaluations on the MMLU dataset (MCQA) utilizing both LLaMA-3-8B-Instruct and LLaMA-3.1-8B-Instruct demonstrate that TRON is more predictive efficient than LAC (Table 3). ***(Corresponding to Weakness 4)***
>
> Overall, we hope that these changes have addressed your concerns. We would be grateful for the opportunity to engage further with you to discuss any remaining questions or concerns you may have.

---

> ### Author Response · Authors · 2024-11-21
> **Supplementary Response to Weakness 1**
>
> For Weakness 1, we test two additional methods to evaluate the semantic equivalence between different responses: (1) Rouge-L and (2) SentenceTransformers (ST) utilizing DistillRoBERTa as the backbone [1][2]. Rouge-L deems two response as semantically equivalent if their longest common subsequence is larger than a threshold. DistillRoBERTa outputs the semantic similarity score between two responses. Following prior work [1][2][3], we adopt two thresholds: 0.5 and 0.7, for both metrics.
>
> Then, we compare bidirectional entailment (BE) based on DeBERTa-large-mnli with the two methods. Since these three methods all operate in the second phase of TRON (i.e., identify), we set alpha to 0 and employ calibration and test data points that cover acceptable responses in their candidate sets. We evaluate the empirical miscoverage rate (EMR) when beta is set to a strict value, 0.1. Results on the VMME dataset utilizing the Gemini-1.5-Pro model are shown below:
>
> | Methods | BE     | Rouge-L (0.5) | Rouge-L (0.7) | ST (0.5) | ST (0.7) |
> |---------|--------|---------------|---------------|----------|----------|
> | EMR     | 0.0847 | 0.0870        | 0.0902        | 0.0829   | 0.0848   |
>
> The guarantees of the second phase of TRON were not compromised by the changes in the semantic equivalence evaluation methods.
>
> Furthermore, in work [4], bidirectional entailment based on DeBERTa-large-mnli is also used within the conformal prediction framework.
>
> In the last paragraph of Appendix C, we have provided a detailed explanation of using bidirectional entailment to determine semantic equivalence.
>
> We would like to know if we have addressed your comments, or if there is anything else we can help to clarify. Thank you again for your helpful comments, and for taking the time to review our work.
>
> ---
>
> ### References
>
> [1] Shifting Attention to Relevance: Towards the Predictive Uncertainty Quantification of Free-Form Large Language Models (ACL 2024)
>
> [2] Word-Sequence Entropy: Towards Uncertainty Estimation in Free-Form Medical Question Answering Applications and Beyond (Engineering Applications of Artificial Intelligence, 2024)
>
> [3] Generating with Confidence: Uncertainty Quantification for Black-box Large Language Models (TMLR 2024)
>
> [4] Addressing Uncertainty in LLMs to Enhance Reliability in Generative AI (NeurIPS 2024 Workshop SafeGenAi)

---

> ### Author Response · Authors · 2024-11-25
> **Have we addressed your concern?**
>
> Dear Reviewer Tw6D,
>
> Thank you again for taking the time to review our paper and providing detailed feedback. As the end of the discussion period is approaching, we want to follow up to see if you have any additional questions we have not addressed, or if there is anything else we can help clarify. We have tried responding to the comments from your initial review, and we are more than happy to discuss any points further. Thank you!
>
> Authors of paper 4201

---

> > ### Comment · Reviewer_Tw6D · 2024-11-26
> >
> > I thank the authors for addressing all my concerns. I have increased my score.

---

> > > ### Author Response · Authors · 2024-11-27
> > > **Appreciation for Feedback and Recognition**
> > >
> > > Thank you for taking the time to review our responses. We appreciate your valuable feedback and constructive insights throughout this process. Thanks again for your recognition.
> > >
> > > Best regards,
> > >
> > > Authors of Paper 4201

---

> ### Author Response · Authors · 2024-11-26
>
> Dear Reviewer Tw6D,
>
> Thank you again for your insightful comments. This is just a gentle reminder as the ICLR discussion period is extended. Could you please take a look at our rebuttal and other reviews, and see whether you would like to update your ratings? We would like to respond to any remaining questions or concerns you may have. Thank you!
>
> Authors of paper 4201

---

### Official Review · Reviewer_rC7h · 2024-11-04

**Soundness:** 3
**Presentation:** 3
**Contribution:** 3
**Rating:** 6
**Confidence:** 3

**Summary:**

The authors propose a two-step framework, TRON, for assessing and controlling risk in MLLMs, specifically targeting VideoQA tasks. The framework consists of: (1) Sampling Step: This step involves calibrating a conformal score that defines the minimum number of response samples needed to ensure the inclusion of acceptable responses within a certain risk level. (2) Identification Step: In this phase, a nonconformity score based on self-consistency theory identifies high-quality responses within the candidate set. This score accounts for model uncertainty by estimating reliability via response frequency, introducing another risk level to statistically control errors.

The authors carry sufficient experiments on four VideoQA datasets with eight MLLMs and demonstrate that TRON achieves desired error rates within the specified risk bounds. The study further highlights TRON’s adaptability and stability through deduplicated prediction sets that provide more efficient and robust uncertainty estimation across various risk levels.

The authors address limitations in existing SCP methods, which either modify outputs to ensure factuality, rely on internal token sequence logits, or restrict applications to multiple-choice settings. This new approach is versatile, applicable to both open-ended and closed-ended VideoQA tasks, and operates independently of internal model outputs.

**Strengths:**

1. TRON’s two-step approach combines conformal scores and self-consistency theory to establish a flexible and robust risk assessment framework for MLLMs, particularly in open-ended scenarios, where traditional SCP methods fall short.
2. The paper presents extensive experiments across four VideoQA datasets and eight MLLMs, showcasing TRON's effectiveness and consistency in different VideoQA tasks.
3. By avoiding reliance on model logits, TRON is adaptable for API-restricted MLLMs, expanding its usability in various practical applications.

**Weaknesses:**

I raise the concern that although TRON is evaluated on diverse datasets, it primarily focuses on VideoQA tasks. Could it be tested on additional multimodal tasks to enhance the generalizability of its risk management capabilities?

**Questions:**

1. How does TRON handle outliers in response sampling that may disproportionately affect the frequency-based confidence scoring?
2. Could TRON’s conformal score be further adapted to dynamically adjust the sampling size based on real-time uncertainty measurements?

---

> ### Author Response · Authors · 2024-11-15
> **Responses to Reviewer rC7h Comments on Weakness and Questions**
>
> > **Response to the weakness:**
>
> Thank you for your valuable insight. TRON is applicable to various generative language models and tasks as it formulates the criterion by analyzing the output distribution of the generative model. We have evaluated TRON on the **open-book conversational QA** dataset, CoQA [1], and the **closed-book reading comprehension** dataset, TriviaQA [2], utilizing the large language models (LLMs), LLaMA-3-70B-Instruct and LLaMA-3.1-70B-Instruct. Additionally, we employ PandaGPT-13B and GPT-4o on the Visual Question Answering (VQA) dataset [3] for **image understanding** tasks. Empirical evaluations have been added to Appendix H (Figure 8 and 9) in the rebuttal revision. The results demonstrate that TRON can also provide risk management for both MLLMs and LLMs on image understanding, conversational QA, and reading comprehension tasks.
>
> > **Response to question 1:**
>
> **Firstly**, since we randomly assign the calibration and test datasets, any sampling anomalies are expected to be uniformly distributed across both sets. **Secondly**, we use the same number of samples for both the calibration and test data. While sampling outliers may be present, the nonconformity score is inherently linked to the correct response. Given that the test and calibration datasets are exchangeable, the overall correctness coverage remains unaffected. However, this may result in an increase in the final average set size (prediction efficiency). **Additionally**, there will be some queries for which we cannot obtain a correct response no matter how many times we sample. We identify these as a distribution shift problem and exclude these queries. **Furthermore**, since we can derive the minimum number of samples on the calibration set at a user-accepted risk level, we can, as prior knowledge, appropriately increase the sampling frequency on the independent and identically distributed (i.i.d.) test data to improve the accuracy of the frequency-based confidence scoring. **For example, at a certain risk level $\alpha$, we determine that the minimum number of samples required is $M$. At this point, we can appropriately increase the number of samples beyond $M$ to enhance the representational capability of frequency-based confidence scoring within the cost constraints, while maintaining the guarantee of the first stage (i.e., $\leqslant \alpha$).**
>
> > **Response to question 2:**
>
> It is feasible under the condition of exchangeability. Based on real-time uncertainty measurements, we can exclude problems that the language model is unable to solve. Then we can set the conformal score of each calibration data point to be any number of samples that ensures the inclusion of the correct answer. At this point, we can set the number of samples for the test data to the maximum number of samples in the calibration set, which ensures that we will definitely sample a correct answer under exchangeable conditions. Then, we can use reliability assessment methods to identify high-quality responses.
>
> Additionally, we can perform real-time uncertainty estimation within the candidate set while sampling. We can define the conformal score to be $r(X_i, Y_i)= inf\lbrace  M_i : Y_i \in \lbrace \hat y_{m}^{(i)}  \rbrace_{m=1}^{M_i}, U(\lbrace \hat y_{m}^{(i)}  \rbrace_{m=1}^{M_i})=0 \rbrace,$ where the $U$ function value is 0, when the uncertainty in the candidate set is below the user's requirement; otherwise, it is 1. However, this depends on the performance of the uncertainty assessment method.
>
> Furthermore, We can evaluate the uncertainty in the candidate set while sampling. When the uncertainty is below a certain threshold, we check whether the sampled response is correct, thus inferring the minimum confidence level or maximum uncertainty that allows for the inclusion of a correct response in the candidate set.
>
> Thank you for your valuable suggestions to improve the paper. We would like to provide responses if we have misunderstood any points of your questions.
>
>
> ---
>
> ### References
>
> [1] Reddy S, Chen D, Manning C D. Coqa: A conversational question answering challenge[J]. Transactions of the Association for Computational Linguistics, 2019, 7: 249-266.
>
> [2] Joshi M, Choi E, Weld D S, et al. TriviaQA: A Large Scale Distantly Supervised Challenge Dataset for Reading Comprehension[C]//Proceedings of the 55th Annual Meeting of the Association for Computational Linguistics. 2017: 1601-1611.
>
> [3] Goyal Y, Khot T, Summers-Stay D, et al. Making the v in vqa matter: Elevating the role of image understanding in visual question answering[C]//Proceedings of the IEEE conference on computer vision and pattern recognition. 2017: 6904-6913.

---

> ### Author Response · Authors · 2024-11-21
> **Rebuttal Revision**
>
> We would like to thank you for your detailed and considerate reviews of our paper. We have uploaded a modified version of our paper that incorporates your comments.
>
> - In the main text, we add a conditional performance analysis of TRON utilizing the size-stratified miscoverage rate in Section 4.4, Paragraph Conditional Miscoverage Rate (Figure 6) for the applicability in real-world critical applications requiring more stringent guarantees. Empirical results demonstrate that the miscoverage rate varies at different set sizes, and the conditional performance of TRON significantly improves by integrating the semantic similarity information into the reliability measurement (e.g., the maximum miscoverage rate decreases from 0.2065 to **0.1667** on the VMME dataset when the upper bound is set to 0.19).
>
> - In Appendix A, we add a brief illustration to conformal prediction utilizing classification tasks.
>
> - In Appendix H, we evaluate the generalizability of TRON to other generative language models in various open-ended tasks. We consider the **(1) open-book conversational QA** dataset, CoQA, and the **(2) closed-book reading comprehension** dataset, TriviaQA, utilizing the large language models (LLMs), LLaMA-3-70B-Instruct and LLaMA-3.1-70B-Instruct (from LangChain DeepInfra). Additionally, we employ PandaGPT-13B and GPT-4o on the Vision Question Answering (VQA) dataset for **(3) image understanding** tasks. Empirical results demonstrate that TRON can also provide risk management for both MLLMs and LLMs on image understanding, conversational QA, and reading comprehension tasks (Figures 8 and 9). ***(Corresponding to Weakness 1)***
>
> - Furthermore, we compare TRON with Least Ambiguous set-valued Classifiers (LAC) in closed-ended settings, which has been proven to produce prediction sets with the smallest average size. Evaluations on the MMLU dataset (MCQA) utilizing both LLaMA-3-8B-Instruct and LLaMA-3.1-8B-Instruct demonstrate that TRON is more predictive efficient than LAC (Table 3).
>
> Overall, we hope that these changes have addressed your concerns. We would be grateful for the opportunity to engage further with you to discuss any remaining questions or concerns you may have.

---

> ### Author Response · Authors · 2024-11-22
> **Supplementary Response to Question 2**
>
> > **Question 2:** Could TRON’s conformal score be further adapted to dynamically adjust the sampling size based on real-time uncertainty measurements?
>
> **Insights:** We envision a possible approach for estimating the overall uncertainty in the candidate set (i.e., sampled responses) using a certain uncertainty measure, and then derive a conformal uncertainty criterion by associating it with the correct response.
>
> For example, employing semantic entropy (SE) [1][2] or shifting attention to relevance (SAR) [3] as the uncertainty measurement $U$(·), we estimate the reliability of the current question-answering by sampling multiple (i.e.,$M$) responses and incorporating their semantic uncertainty. ***Formally***, we define the uncertainty function as $U(x_i,\lbrace  y_{j}  \rbrace_{j=1}^{M_i} )$, where $M_i$ denotes the sampling size for i-th question $x_i$, and $y_j$ denotes the j-th sampled response within the candidate set. ***Then***, we can utilize several validation data points to obtain an uncertainty interval function $A(M)$, which represents the **empirical score range** within which the value of $U(x,\lbrace  y_{j} \rbrace_{j=1}^{M} )$ should fall, when the size of the candidate set is $M$ and $\lbrace  y_{j} \rbrace_{j=1}^{M}$ covers the ground-truth answer. ***Finally***, we can define the conformal score of each calibration data point as $s_i=inf\lbrace M_i : U(x_i,\lbrace  y_{j} \rbrace_{j=1}^{M_i} )∈ A(M_i)\rbrace$ and at this point, **the conformal score can dynamically adjust the sampling size until the overall uncertainty of the candidate set falls within our pre-defined uncertainty interval $A(M_i)$**. Considering the sampling cost, we select the minimum $M_i$ (i.e., $inf$). ***Furthermore***, we can also employ the calibration set to assess the reliability of function $A$(·) at various sampling size M and calculate the corresponding error rate
> $\delta(M_i) = 1 - \mathbb{P}(\lbrace U(x_i,\lbrace  y_{j} \rbrace_{j=1}^{M_i})∈ A(M_i)\rbrace \equiv \lbrace y^* ∈ \lbrace  y_{j} \rbrace_{j=1}^{M_i} \rbrace )$.
> At this point, $s_i$ is miscalibrated at a risk level $\delta$.
>
> Note that this requires the calibration data, validation data, and test data to be independent and identically distributed (i.i.d.), without any distribution shift issues.
>
> We are grateful to you for recognizing the novelty and contribution of our research and providing thoughtful feedback. We would like to discuss any remaining questions or concerns you may have.
>
> ---
>
> ### References
> [1] Semantic uncertainty: Linguistic invariances for uncertainty estimation in natural language generation (ICLR 2023).
>
> [2] Shifting Attention to Relevance: Towards the Predictive Uncertainty Quantification of Free-Form Large Language Models (ACL 2024).

---

> ### Author Response · Authors · 2024-11-25
> **Have we addressed your concern?**
>
> Dear Reviewer rC7h,
>
> Thank you again for taking the time to review our paper and providing detailed feedback. As the end of the discussion period is approaching, we want to follow up to see if you have any additional questions we have not addressed, or if there is anything else we can help clarify. We have tried responding to the comments from your initial review, and we are more than happy to discuss any points further. Thank you!
>
> Authors of paper 4201

---

> > ### Comment · Reviewer_rC7h · 2024-11-25
> > **Reply to Authors' responses**
> >
> > Thanks for the author’s response, which addressed my previous concerns. I have updated my score accordingly.

---

> > > ### Author Response · Authors · 2024-11-25
> > > **Appreciation for Feedback and Recognition**
> > >
> > > Thanks for taking the time to review our responses and raising your score. We are pleased that our responses and revision address your concerns, which also improve the quality of this work.

---

### Official Review · Reviewer_amj3 · 2024-11-05

**Soundness:** 3
**Presentation:** 3
**Contribution:** 3
**Rating:** 8
**Confidence:** 4

**Summary:**

The paper introduces a two step risk control based framework extending split conformal prediction method for open ended and multimodal (videoQA) tasks. The method applies a conformal score to calibrate the minimum number of responses (samples) needed to ensure coverage. This score defines the prediction set’s error rate at a risk level alpha. It then refines the set using frequency or semantic similarity to identify high quality responses controlled by another risk parameter beta. The overall risk is bounded by a function of alpha and beta to provide statistical consistency guarantees using calibration, sampling and refinement steps. The paper builds upon multiple conformal risk control based methods and addresses the shortcomings with this two step method (to maintain smaller average prediction set size (APSS))  with lower risk thresholds) and applying them to multimodal videoQA setting.

**Strengths:**

- The paper's two-step risk control methodology addresses general shortcoming of the conformal prediction method and provides statistical guarantees for error rates, increasing the reliability of MLLM responses.
- Since open ended tasks are more challenging due to large number of possible generation, this method (two step approach, use of semantic similarity) seems to dynamically adapt well to provide flexible prediction set sizes for complex and generic generative scenarios (although we need more experimental validation). Error stays within bounds even after filtering step (bounded by alpha + beta - alpha.beta)
- As shown is Figure 3, deduplication of the semantically similar responses helps with more stable error rates and smaller prediction sets. Experiments suggest that semantic diversity can create smaller, more efficient prediction sets (lower APSS) without compromising on accuracy (EER stays within limits).

**Weaknesses:**

- Authors already mention under limitations that guarantees are not conditional to individual data points but marginal over the test set. With this limitation, it may still be a bottleneck where risk compliance guarantees are needed for critical applications requiring more stringent guarantees and/or compliance requirements.
- more open-ended evaluations and experiments on the open-ended datasets would have shed more light on the strengths and weaknesses of the methods (like Fig 4b). This is a key innovative strength of the method to address open-ended tasks (unlike MCQ) and adoption of this method will depend heavily on understand the strengths of this method in more open-ended generation tasks.

**Questions:**

- Do you foresee any major modifications needed for TRON to control risk in scenarios involving distribution shifts, where calibration and test distributions differ?
- Would it be feasible to incorporate dynamic adjustments to prediction set sizes based on task difficulty or user preferences in real time? Are there challenges with balancing efficiency and robustness in such a dynamic setting?
- Could access to model internals like logits help improve TRON's performance?
- Could reliance on frequency-based nonconformity scores lead to biases in the types of responses included in the prediction set, especially in cases where the model’s sampling is limited?
- Have you observed any variance in EER across different model architectures or response generation and sampling methods (for example, cases where EER can go outside the bounds set by alpha and beta)
- How easy is it to generalize this approach beyond VideoQA to other open-ended tasks? Are there any major limitations or requirements for generalization?

---

> ### Author Response · Authors · 2024-11-14
> **Responses to Reviewer amj3 Comments on Two Weaknesses**
>
> > Weakness 1: Authors already mention under limitations that guarantees are not conditional to individual data points but marginal over the test set. With this limitation, it may still be a bottleneck where risk compliance guarantees are needed for critical applications requiring more stringent guarantees and/or compliance requirements.
>
> ***Response:*** Thank you for pointing out the bottleneck in risk control. Conditional coverage is a stronger property than marginal coverage, and in the most general case, conditional coverage is impossible to achieve [1][2][3]. We have checked how close our method comes to approximating it utilizing the size-stratified coverage metric (i.e., the stratified coverage at each size of prediction set) [2][4][5]. We utilize the Gemini-1.5-Pro model and set $\alpha = \beta = 0.1$. Empirical evaluations on four datasets indicate that the set size ranges from 0 to 3 after semantically deduplication. The results of size-stratified miscoverage rate are shown in the table below. **We have added an evaluation of conditional performance in the rebuttal revision (Paragraph Conditional Miscoverage Rate and Figure 6 in Section 4.4)**.
>
> | **Dataset\Set Size** | **1**    | **2**    | **3**    |
> |:---------------------:|:--------:|:--------:|:--------:|
> | **VMME**             | 0.1788   | 0.1531   | 0.2065   |
> | **NEXT**             | 0.0900   | 0.1778   | 0.2244   |
> | **MUSC**             | 0.1250   | 0.2286   | 0.1830   |
> | **MSVD**             | 0.0778   | 0.1222   | 0.1556   |
>
> > Weakness 2: more open-ended evaluations and experiments on the open-ended datasets would have shed more light on the strengths and weaknesses of the methods (like Fig 4b). This is a key innovative strength of the method to address open-ended tasks (unlike MCQ) and adoption of this method will depend heavily on understand the strengths of this method in more open-ended generation tasks.
>
> ***Response:*** Thank you for your valuable suggestions for improvement. The key innovation of our method is the development of a conformity score that calibrates the number of samples, which approximates the candidate set as multiple-choice options in closed-ended settings at a user-specified risk level for the first time, addressing the issues of external significance level and sensitivity in prior studies [3][5]. Then, in the second step, based on self-consistency, we use frequency to replace logits and establish the prediction set. Observing the semantic redundancy in the prediction set in open-ended settings, we conduct deduplication and employ the calibrated set size for uncertainty estimation. As shown in Figure 4(b), we utilize the average set size to evaluate the uncertainty of MLLM when providing audio modality information at different levels of silence, to supplement the accuracy metric.
>
> **As we stated in the conclusion, our method is applicable to all generative language models and tasks, and we will evaluate our method on both vision-language models (VLMs) and large language models (LLMs). Due to the slow pace of open-ended language generation tasks, we will update the experimental setup and results to the rebuttal revision as soon as possible and notify you through official comment.**
>
> We hope these kindly address the reviewer's concerns. If there are any aspects we have overlooked or misunderstood, please let us know. We would like to provide responses if the reviewer has further questions.
>
> ---
>
> ### References
> [1] Vladimir Vovk. 2012. Conditional Validity of Inductive Conformal Predictors. In Proceedings of the Asian Conference on Machine Learning, PMLR.
>
> [2] Anastasios N Angelopoulos and Stephen Bates. 2021. A gentle introduction to conformal prediction and distribution-free uncertainty quantification. arXiv preprint arXiv:2107.07511.
>
> [3] Victor Quach, Adam Fisch, Tal Schuster, Adam Yala, Jae Ho Sohn, Tommi S. Jaakkola, and Regina Barzilay. 2024. Conformal language modeling. In The Twelfth International Conference on Learning Representations.
>
> [4] Bhawesh Kumar, Charles Lu, Gauri Gupta, Anil Palepu, David Bellamy, Ramesh Raskar, and Andrew Beam. 2023. Conformal prediction with large language models for multi-choice question answering. arXiv preprint arXiv:2305.18404.
>
> [5] Jiayuan Su, Jing Luo, Hongwei Wang, Lu Cheng. 2024. API Is Enough: Conformal Prediction for Large Language Models Without Logit-Access. In Findings of the Association for Computational Linguistics: EMNLP 2024.

---

> ### Author Response · Authors · 2024-11-14
> **Responses to Reviewer amj3 Comments on Six Questions**
>
> > **Response to question 1:**
>
> Theoretically, distribution shift is primarily due to the quantile of nonconformity scores obtained from the calibration set being shifted with respect to the test distribution. In our view, distribution shift in language generation tasks is attributed to the conditional nature of language models. For example, It is possible that for a powerful proprietary model, the difficulty of dataset A and dataset B is similar. In this case, we can consider that datasets A and B satisfy the exchangeability condition for that model. However, if a language model is only suitable for the types of questions in dataset A and struggles with questions in dataset B, a distribution shift phenomenon will occur. We believe that distribution shift is determined by the language model itself, and we can achieve alignment across various distributions by weighting the nonconformity scores based on the analysis of certain model statistical metrics. We are currently working on this.
>
> > **Response to question 2:**
>
> We can retrieve specific types of calibration data based on user needs, thereby determining the quantile threshold by constraining the size of the prediction set on the calibration set to handle the test data. In this dynamic environment, the criteria for selecting calibration data are particularly important, as different evaluation criteria can lead to deviations in the exchangeability between calibration data and test data. At this point, the efficiency of data filtering and the robustness of risk control need to be balanced according to actual requirements.
>
> > **Response to question 3:**
>
> Yes. Since the nonconformity score is defined to reflect how the response disagrees with the question, the evaluation of response reliability will impact the final results of risk control. As we mentioned in the last paragraph in Section 3.2 (i.e., Extensibility), the function F(·) can be any measure that reflects the trustworthiness of each response. Additionally, Table 2 in Section 4.4 shows that when we incorporate semantic diversity into  F(·), the average set size decreases (i.e., more efficient prediction). Relying solely on black-box methods to evaluate responses is limited, and integrating auxiliary information, such as logits-based entropy, can enhance the reliability assessment and thereby improve TRON's performance.
>
> Inspired by the weakness 1 you mentioned, **we have added a comparison of the conditional performance of two measures in Section 4.4**. The table below shows that after incorporating semantic similarity information, TRON's conditional performance has significantly improved (e.g., **the size-stratified miscoverage rate decreases from 0.2065 to 0.1667 on the VMME dataset when the upper bound is 0.19**).
>
> |          | Frequency   |   |  | Semantic  |Diversity                 |                 |
> |----------|---------------------|-----------------|-----------------|----------------------|-----------------|-----------------|
> | **Set Size** | 1  | 2  | 3 | 1  | 2   | 3 |
> | **VMME** | 0.1788 | 0.1531 | 0.2065 | 0.1203 | 0.1667 | 0.1389  |
> | **NEXT** | 0.0900 | 0.1778 | 0.2244 | 0.1472 | 0.1763 | 0.2075|
> | **MUSC** | 0.1250  | 0.2286| 0.1830 | 0.1225 | 0.1970 | 0.1923  |
> | **MSVD** | 0.0778| 0.1222 | 0.1556 | 0.1246 | 0.1592 | 0.1875  |
>
> > **Response to question 4:**
>
> When the number of samples is limited, the capability of frequency to represent response confidence will decline, as we mentioned in the last paragraph of Section 3.2 (i.e., Extensibility), which will affect the average set size (i.e., efficiency). At this point, we need to incorporate other auxiliary tools, such as external models, into the function F(·) to enhance the assessment of response reliability. However, theoretically, the risk control of the method remains guaranteed because the number of samples is consistent across all calibration and test data.
>
> > **Response to question 5:**
>
> Yes, while the theoretical guarantee of TRON is rigorous, there can be minor fluctuations in practice due to finite-sample variability [1][2].
>
> > **Response to question 6:**
>
> As we mentioned in the conclusion, our framework can be directly applied to various generative language models and tasks. TRON inherits the model-agnostic property of conformal prediction, and we establish task-specific guarantees solely by analyzing the output distribution of the model.
>
> Thank you very much for your valuable questions. Throughout the rebuttal process, we gained deeper insights into improving the quality of the paper and conducting future work. We would like to provide responses if you have any further questions.
>
> ---
>
> ### References
> [1] Angelopoulos A N, Bates S. A gentle introduction to conformal prediction and distribution-free uncertainty quantification[J]. arXiv preprint arXiv:2107.07511, 2021.
>
> [2] Ye F, Yang M, Pang J, et al. Benchmarking llms via uncertainty quantification[J]. arXiv preprint arXiv:2401.12794, 2024.

---

> ### Author Response · Authors · 2024-11-21
> **Rebuttal Revision**
>
> We would like to thank you for your detailed and considerate reviews of our paper. We have uploaded a modified version of our paper that incorporates your comments.
>
> - In the main text, we add a conditional performance analysis of TRON utilizing the size-stratified miscoverage rate in Section 4.4, Paragraph Conditional Miscoverage Rate (Figure 6) for the applicability in real-world critical applications requiring more stringent guarantees. Empirical results demonstrate that the miscoverage rate varies at different set sizes, and the conditional performance of TRON significantly improves by integrating the semantic similarity information into the reliability measurement (e.g., the maximum miscoverage rate decreases from 0.2065 to **0.1667** on the VMME dataset when the upper bound is set to 0.19). ***(Corresponding to Weakness 1 and Question 3)***
>
> - In Appendix A, we add a brief illustration to conformal prediction utilizing classification tasks.
>
> - In Appendix H, we evaluate the generalizability of TRON to other generative language models in various open-ended tasks. We consider the **(1) open-book conversational QA** dataset, CoQA, and the **(2) closed-book reading comprehension** dataset, TriviaQA, utilizing the large language models (LLMs), LLaMA-3-70B-Instruct and LLaMA-3.1-70B-Instruct (from LangChain DeepInfra). Additionally, we employ PandaGPT-13B and GPT-4o on the Vision Question Answering (VQA) dataset for **(3) image understanding** tasks. Empirical results demonstrate that TRON can also provide risk management for both MLLMs and LLMs on image understanding, conversational QA, and reading comprehension tasks (Figures 8 and 9). ***(Corresponding to Weakness 2 and Question 6)***
>
> - Furthermore, we compare TRON with Least Ambiguous set-valued Classifiers (LAC) in closed-ended settings, which has been proven to produce prediction sets with the smallest average size. Evaluations on the MMLU dataset (MCQA) utilizing both LLaMA-3-8B-Instruct and LLaMA-3.1-8B-Instruct demonstrate that TRON is more prediction-efficient than LAC (Table 3).
>
> Overall, we hope that these changes have addressed your concerns. We would be grateful for the opportunity to engage further with you to discuss any remaining questions or concerns you may have.

---

> ### Author Response · Authors · 2024-11-23
> **Supplementary Response to Question 2**
>
> > Question 2: Would it be feasible to incorporate dynamic adjustments to prediction set sizes based on task difficulty or user preferences in real time? Are there challenges with balancing efficiency and robustness in such a dynamic setting?
>
> **Our insights:** We envision a possible approach, which estimates the overall uncertainty in the candidate set (i.e., sampled responses) using a certain uncertainty measure and then derives a conformal uncertainty criterion by associating the uncertainty condition with the correct response.
>
> For example, employing semantic entropy (SE) [1][2] or shifting attention to relevance (SAR) [3] as the uncertainty measurement $U$(·), we estimate the reliability of the current question-answering by sampling multiple (i.e.,$M$) responses and incorporating their semantic uncertainty. ***Formally***, we define the uncertainty function as $U(x_i,\lbrace  y_{j}  \rbrace_{j=1}^{M_i} )$, where $M_i$ denotes the sampling size for the i-th question $x_i$, and $y_j$ denotes the j-th sampled response within the candidate set. ***Then***, we can utilize a validation data set to obtain an uncertainty interval function $A(M)$, which represents the **empirical score range** within which the value of $U(x,\lbrace  y_{j} \rbrace_{j=1}^{M} )$ should fall, when the size of the candidate set is $M$ and $\lbrace  y_{j} \rbrace_{j=1}^{M}$ covers the ground-truth answer. ***Finally***, we can define the conformal score of each calibration data point as $s_i=inf\lbrace M_i : U(x_i,\lbrace  y_{j} \rbrace_{j=1}^{M_i} )∈ A(M_i)\rbrace$ and at this point, **the conformal score can dynamically adjust the sampling size until the overall uncertainty of the candidate set falls within our pre-defined uncertainty interval $A(M_i)$**. Considering the sampling cost (or efficiency), we select the minimum $M_i$ (i.e., $inf$). ***Furthermore***, we can also employ the calibration set to assess the reliability of function $A$(·) at various sampling size $M_i$ and calculate the corresponding error rate
> $\delta(M_i) = 1 - \mathbb{P}(\lbrace U(x_i,\lbrace  y_{j} \rbrace_{j=1}^{M_i})∈ A(M_i)\rbrace \equiv \lbrace y^* ∈ \lbrace  y_{j} \rbrace_{j=1}^{M_i} \rbrace )$.
> At this point, $s_i$ is miscalibrated at a risk level $\delta$.
>
> Note that this requires all calibration, validation, and test data points to be independent and identically distributed (i.i.d.), without any distribution shift issues.
>
> We are grateful to you for recognizing the novelty and contribution of our research and providing thoughtful feedback. We would like to discuss any remaining questions or concerns you may have.
>
> ---
>
> ### References
> [1] Semantic uncertainty: Linguistic invariances for uncertainty estimation in natural language generation (ICLR 2023).
>
> [2] Shifting Attention to Relevance: Towards the Predictive Uncertainty Quantification of Free-Form Large Language Models (ACL 2024).

---

> ### Author Response · Authors · 2024-11-27
> **Appreciation for Feedback and Recognition**
>
> Thank you for taking the time to review our paper. We appreciate your valuable feedback and constructive insights throughout this process. We are very grateful for your recognition of our work and thorough suggestions for improvements in our future work.
>
> Best regards,
>
> Authors of Paper 4201

---

### Author Response · Authors · 2024-11-23
**Summary of Author Response to All the Reviewers**

We would like to thank all the reviewers for their insightful comments. We revised our paper based on the constructive feedback and suggestions from the reviewers. We marked the contents that already existed in the original submission (but may be missed by reviewers) in red, and those revised or newly added contents in blue in the revision. Our key responses are summarized as follows:

**Additional explanations.**

- As Reviewer Tw6D suggested, we explained the Silence Percentage (SP) conducted on the audio modality information in Section 4.3. In addition, we analyzed why it is necessary to assess the uncertainty based on the average prediction set size (APSS), in order to assist accuracy in the comprehensive evaluation of MLLMs. Moreover, we detailed the semantic equivalence method in Appendix C.

- As Reviewer pRzK suggested, we illustrated the base framework of split conformal prediction utilizing classification tasks in Appendix A.  In addition, we explained the evaluation metrics (e.g., APSS) in Appendix F.

**Additional experimental results.**

- As all four reviewers suggested, we generalized our framework to other models (e.g., large language models and vision-language models) on additional open-ended tasks (e.g., conversational QA, reading comprehension, image understanding) in Appendix H.

- As Reviewers amj3 and pRzK suggested, we evaluated the conditional performance of our framework for real-world critical applications requiring more stringent guarantees in Section 4.4. In addition, we discussed improving the conditional performance by utilizing more comprehensive reliability measures of model generations.

- As Reviewers Tw6D and pRzK suggested, we compared our framework with existing methods in close-ended scenarios in Appendix H.

**Additional insights.**

- As Reviewer amj3 suggested, we discussed the future work for adapting our framework to language generation tasks under distribution shift.

- As Reviewer rC7h suggested, we discussed how does our framework handle outliers in sampling size.

- As both Reviewers amj3 and rC7h suggested, we envisioned a feasible approach, which further adapts our framework to dynamically adjust the sampling size based on real-time uncertainty measurements.

- As Reviewer Tw6D suggested, we analyzed how to determine the sample ratio for the calibration set and test set based on different models.

- As Reviewer pRzK suggested, we provided an example of medical video analysis, attempting to validate the application of our method in real-world or industry-specific VideoQA tasks.

We thank all the reviewers again for the detailed and constructive review. We are pleased to see the reviewers' acknowledgment of the contribution of the proposed method. Most of the concerns are raised about unclear expressions, generalizability, and future work. We hope our explanation, additional experimental results, and insights for further adaptation in the rebuttal could address all of your concerns. Please let us know if you have any questions or concerns.

---

### Meta-Review · Area_Chair_uhBp · 2024-12-16

**Metareview:**

This paper presents a novel pipeline for risk assessment for open-ended LLM generated responses to visual question answering systems.  This work addresses limitations of the current state of the art method of using Split Conformal Prediction (SCP) to construct estimates of the error rate of statistical prediction methods for LLM assessment.

Extracting a well written summary from one of the reviewers: "The authors propose a two-step framework, TRON, for assessing and controlling risk in MLLMs, specifically targeting VideoQA tasks. The framework consists of: (1) Sampling Step: This step involves calibrating a conformal score that defines the minimum number of response samples needed to ensure the inclusion of acceptable responses within a certain risk level. (2) Identification Step: In this phase, a nonconformity score based on self-consistency theory identifies high-quality responses within the candidate set. This score accounts for model uncertainty by estimating reliability via response frequency, introducing another risk level to statistically control errors.

The authors carry sufficient experiments on four VideoQA datasets with eight MLLMs and demonstrate that TRON achieves desired error rates within the specified risk bounds. The study further highlights TRON’s adaptability and stability through deduplicated prediction sets that provide more efficient and robust uncertainty estimation across various risk levels.

The authors address limitations in existing SCP methods, which either modify outputs to ensure factuality, rely on internal token sequence logits, or restrict applications to multiple-choice settings. This new approach is versatile, applicable to both open-ended and closed-ended VideoQA tasks, and operates independently of internal model outputs.

Reading this myself, I agree with the  major points the reviewers have identified.  Many weaknesses are discussed, but the problem of estimating the error of a predictor under and unknown distribution shift is difficult and I believe that this paper both takes us a notable step closer to solving the problem and makes clear its limitations.

**Additional Comments On Reviewer Discussion:**

The reviewer discussion was quite engaged during the author rebuttal period and I was well pleased with the quality of the engagement from the reviewers.  Two. reviewers increased their scores during the discussion and one remained steadfast in their initial score of "8" - accept good paper.  Possible one of the best rebuttal period discussions I have seen.

---

### Decision · Program_Chairs · 2025-01-22

Accept (Spotlight)